DOI: 10.1038/s41467-018-04920-3　　**OPEN**

# Task-induced brain state manipulation improves prediction of individual traits

Abigail S. Greene [ID] [1], Siyuan Gao [ID] [2], Dustin Scheinost[3] & R. Todd Constable [ID] [1,3,4]

Recent work has begun to relate individual differences in brain functional organization to human behaviors and cognition, but the best brain state to reveal such relationships remains an open question. In two large, independent data sets, we here show that cognitive tasks amplify trait-relevant individual differences in patterns of functional connectivity, such that predictive models built from task fMRI data outperform models built from resting-state fMRI data. Further, certain tasks consistently yield better predictions of fluid intelligence than others, and the task that generates the best-performing models varies by sex. By considering task-induced brain state and sex, the best-performing model explains over 20% of the variance in fluid intelligence scores, as compared to <6% of variance explained by rest-based models. This suggests that identifying and inducing the right brain state in a given group can better reveal brain-behavior relationships, motivating a paradigm shift from rest- to task-based functional connectivity analyses.

---

[1] Interdepartmental Neuroscience Program, Yale School of Medicine, New Haven, 06520 CT, USA. [2] Department of Biomedical Engineering, Yale School of Engineering and Applied Science, New Haven, 06520 CT, USA. [3] Department of Radiology and Biomedical Imaging, Yale School of Medicine, New Haven, 06520 CT, USA. [4] Department of Neurosurgery, Yale School of Medicine, New Haven, 06520 CT, USA. Correspondence and requests for materials should be addressed to A.S.G. (email: abigail.greene@yale.edu)

The potential to "decode" brain activity has generated much excitement in recent years[1,2], both because such analyses offer a window into neural representation[3], and because, by exploring individual differences in these representations, we can characterize, predict, and ultimately alter brain–behavior relationships in health and disease[4,5]. Functional magnetic resonance imaging (fMRI) provides a means to pursue these goals[1,4], particularly given the shift from studying circumscribed brain regions to leveraging whole-brain, data-driven techniques to explore distributed patterns of activation[3] and connectivity[6,7]. This transition reflects a growing consensus that important insights into neural function may be found in the organization and coordination of distributed circuitry[8,9], making connectome-based approaches well suited for advancing predictive modeling efforts[10,11].

Connectome-based analyses usually focus on resting-state fMRI[6] (hereafter, "rest data" and "rest-based" analyses), but rest is an unconstrained state[12] that may fail to capture the full range of individual differences in functional connectivity[13,14]. Acquiring fMRI data while subjects perform a task (hereafter, "task data" and "task-based" analyses) provides a means to practically and objectively manipulate brain state, and thus to explore its effects on patterns of functional connectivity, individual differences in these patterns, and the relationship of these individual differences to cognition and behavior[15]. Previous applications of connectome-based analyses to task data have focused on characterizing similarities, differences, and transitions among intrinsic and task-induced brain states[16], or on predicting state variables directly related to the task (e.g., attention[17]), but the utility of task-based functional connectivity for prediction of stable, individual traits has been relatively unexplored.

In this work, we test the hypothesis that, much like a cardiac stress test identifies symptoms not observable at rest, tasks may tax individuals along a particular cognitive dimension, thereby amplifying individual differences in underlying neural circuitry and improving predictive models of related cognitive traits. To do so, we apply connectome-based predictive modeling (CPM)[18] to two, independent data sets (Human Connectome Project (HCP)[19] and Philadelphia Neurodevelopmental Cohort (PNC)[20]), and show that models built from task data better predict fluid intelligence (gF) than do those built from rest data, thus extending previous work on task-induced changes in functional connectivity by demonstrating the utility of these changes for prediction of stable traits. Not only do task-based models outperform rest-based models, but certain tasks consistently yield better gF predictions than others, suggesting that some states may be better suited to reveal trait-relevant individual differences in functional organization. Moreover, we demonstrate that states that most improve trait prediction show a sex dependence. Optimizing models for prediction may thus require consideration of factors such as sex in addition to brain state. These findings replicate, and predictive models generalize, across data sets, suggesting the broad relevance of these findings.

Altogether, the results suggest an opportunity to use tasks to perturb the brain during fMRI acquisitions in order to more comprehensively characterize individual differences in the neural circuitry underlying complex traits, and to generate useful behavioral and clinical predictions about individuals on the basis of these differences.

## Results

**Connectome-based predictive modeling.** These analyses used fMRI data from the Human Connectome Project (HCP; $n = 515$); each subject performed 2 rest ("rest1" and "rest2") and 7 task (gambling, language, motor, relational, social, working memory

(WM), and emotion) conditions in the scanner[19]. These data were parcellated into 268 nodes using a whole-brain, functional atlas defined previously in a separate sample[21,22]. Next, the mean time courses of each node pair were correlated and correlation coefficients were Fisher transformed, generating nine connectivity matrices per subject. Given the complexity of these tasks and the data-driven nature of this analysis, we performed CPM[18] on matrices from each condition to generate cross-validated task- and rest-based predictive models of fluid intelligence (gF), as measured by matrix reasoning test scores (hereafter, "Pmat"; see Methods for measurement details), from whole-brain patterns of functional connectivity. Model performance was quantified as the Spearman's correlation between predicted and true gF ($r_s$) or as gF percent variance explained ($100r_s^2$). For main gF CPM analyses (described in State manipulations improve trait predictions), significance was assessed using 1000 iterations of non-parametric permutation testing that accounted for family structure[23,24], and resulting $P$ values were corrected for multiple comparisons using the false discovery rate[25]; for all remaining post-hoc analyses, except as otherwise noted, significance was assessed parametrically, and uncorrected $P$ values are presented. Where applicable, analyses were performed using both rest1 and rest2 data, with comparable results; for clarity, only rest1 results are reported for most post-hoc analyses. We repeated this analysis using fMRI data from the Philadelphia Neurodevelopmental Cohort (PNC; $n = 571$); each subject performed 1 rest and 2 task (emotion and WM) runs in the scanner[20], and data from all 3 conditions were submitted to the pipeline described above.

Each iteration of the CPM pipeline yields two networks: one comprised of edges that are positively correlated with gF ("correlated network" (CN)) and one comprised of edges that are negatively correlated with gF ("anti-correlated network" (AN)). For simplicity and improved interpretability, information in these networks was consolidated by taking the difference between network strength (i.e., summed edge strengths) in the CN and AN, and this combined network strength was used to train and test the models (Methods). All subsequently reported results are for this combined network, except where otherwise noted.

Because the edge-selection thresholds used to generate these networks are inevitably arbitrary, we tested seven different thresholds; model performance was comparable using all tested thresholds (Supplementary Table 1). Except as otherwise noted, all subsequently reported CPM results were generated using an edge-selection threshold of $P < 0.001$, and validation and overlap analyses were performed using a less conservative threshold ($P < 0.01$) to minimize the effects of overfitting and noise introduced by trait-irrelevant differences (e.g., differences in task implementation and subject age across data sets).

**State manipulations improve trait predictions.** All models, with the exception of those built from HCP rest2 data, yielded predictions that trended toward significance (HCP rest1 and relational task, PNC rest; FDR corrected, $q = 0.05 – 0.06$) or were significant (all other conditions; FDR corrected, $q < 0.05$) in both the HCP (Fig. 1a–d) and PNC (Fig. 1e–h) data sets. Specifically, in the HCP data set, the gambling task yielded the best-performing model: $r_s^2 = 12.8\%$ ($P,q < 0.001$). The WM task yielded the second-best model: $r_s^2 = 10.6\%$ ($P,q < 0.003$; Fig. 1b). Rest yielded the worst-performing models: rest1, $r_s^2 = 2.9\%$ ($P,q = 0.06$; Fig. 1d); rest2, $r_s^2 = 0\%$ ($P,q = 0.86$). In the PNC data set, the WM task yielded the best-performing model: $r_s^2 = 12.3\%$ ($P,q < 0.001$; Fig. 1f). The emotion task yielded the second-best model: $r_s^2 = 9.9\%$ ($P,q < 0.005$; Fig. 1g). Rest yielded the worst-performing model: $r_s^2 = 3.9\%$ ($P,q \leq 0.05$; Fig. 1h). In both data

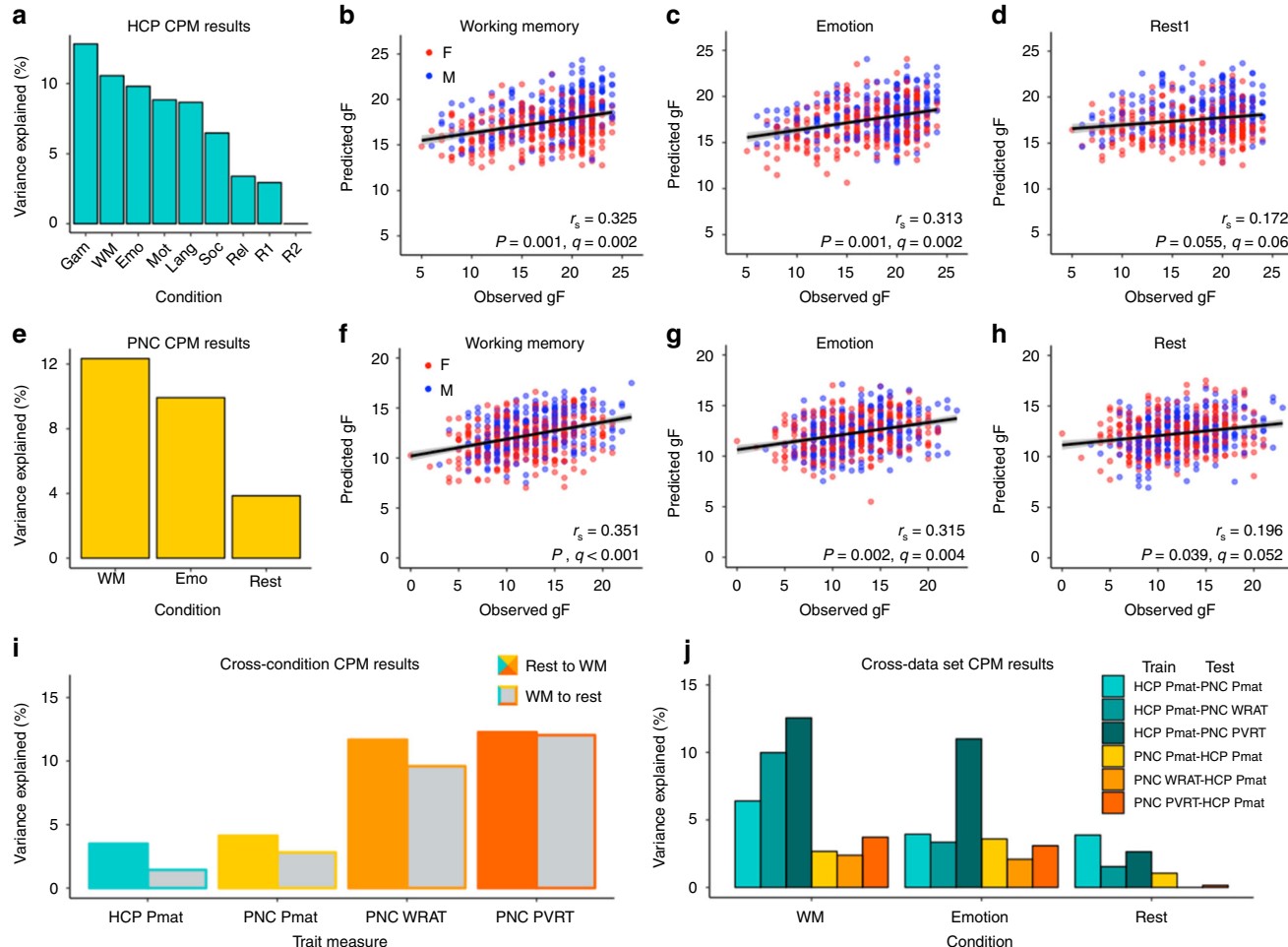

**Fig. 1** Task-induced brain state is a key determinant of individual trait prediction accuracy. **a** Results from the cross-validated CPM pipeline in each of the 9 HCP conditions ($n = 515$) using an edge-selection threshold of $P < 0.001$, plotted as and ordered by percent of fluid intelligence (gF) variance explained. Gam, gambling task; WM, working memory task; Emo, emotion processing task; Mot, motor task; Lang, language task; Soc, social task; Rel, relational task; R1, rest1; R2, rest2. **b–d** Expansion of results presented in **a** for the WM, emotion, and rest1 conditions; each point represents the relationship between predicted and observed gF for a single subject, colored by subject sex (F, female; M, male), plotted with the best-fit line and its 95% CI (gray area). $r_s$, Spearman's correlation coefficient; significance assessed via 1000 iterations of permutation testing. **e–h** Results of CPM analyses in the PNC data set ($n = 571$), presented as in (**a–d**). **i** Results of cross-condition prediction analyses; for each measure, networks built from rest data were applied to WM data ("Rest to WM") and vice versa ("WM to rest") to predict the corresponding measure, using an edge-selection threshold of $P < 0.01$. Pmat, matrix reasoning test of gF; WRAT, Wide Range Achievement Test; PVRT, Penn Verbal Reasoning Test. **j** Results of cross-data set validation analyses. Cool colors indicate HCP-based models; warm colors indicate PNC-based models; shade corresponds to predicted measure in the case of HCP to PNC, and to the measure used for model building in the case of PNC to HCP. In all cases, the same condition was used for model building and prediction, and an edge-selection threshold of $P < 0.01$ was used

sets, some tasks yielded better gF predictions than others, but in all cases, task-based models outperformed rest-based models (rank sum = 71, two-sided $P = 0.018$, Mann–Whitney $U$ test), and this result was stable across 1000 iterations of a split-half prediction analysis (Supplementary Fig. 1).

Moreover, these patterns were not specific to prediction of gF. The same pipeline was applied to the PNC data to predict scores on the Wide Range Achievement Test (WRAT) and the Penn Verbal Reasoning Test (PVRT); prediction accuracy was even higher than for Pmat prediction ($r_s^2 = 8.6$–20.8%, all $P < 2e{-}12$), with task-based models again outperforming rest-based models (Table 1).

**Investigation of potential confounds.** We conducted a number of analyses to confirm the robustness of our results. First, because two versions of the Penn Matrix Reasoning Test were used to

**Table 1 Task-based models outperform rest-based models in prediction of two additional intelligence-related measures (Wide Range Achievement Test (WRAT, $n = 558$) and Penn Verbal Reasoning Test (PVRT, $n = 563$))**

| Task | WRAT | PVRT |
|---|---|---|
| Working memory | 17.9 | 20.8 |
| Emotion identification | 10.4 | 18.5 |
| Rest | 8.6 | 10.2 |

Results reported as percent of WRAT and PVRT score variance explained ($100r_s^2$) by models generated using an edge-selection threshold of $P < 0.001$

measure gF in the PNC, this analysis was repeated to predict percent, rather than number, correct. In addition, the analysis was performed without 11 PNC subjects with less or non-valid Pmat scores, separately for subjects who performed each version, and with incorporation of version into the modeling pipeline at the edge-selection (via partial correlation) and model-building (via multilinear regression) steps (Methods). In all cases, one of the tasks yielded the best-performing model (Supplementary Table 2).

Similarly, given that two different image reconstruction methods were used on the HCP data, CPM was performed after excluding all subjects for whom the r227 algorithm was not available, and with incorporation of reconstruction method into the modeling pipeline at the edge-selection (via partial correlation) and model-building (via multilinear regression) steps. Next, a subset of subjects with identified quality control issues or missing field maps was excluded. As in previous analyses, task-based models outperformed rest-based models in all cases (Supplementary Table 3), suggesting that these issues did not confound main results.

Although we applied strict motion exclusion criteria (Methods) and motion was not correlated with gF in 18 out of 21 runs ($P > 0.05$, Bonferroni corrected), model predictions were frequently correlated with mean frame-to-frame displacement. The CPM pipeline was modified to explicitly control for motion at the edge-selection (via partial correlation) and model-building (via multilinear regression) steps. Neither manipulation substantially affected CPM results (Supplementary Table 4), suggesting that the shared variance between motion and model predictions, and between model predictions and gF is largely non-overlapping.

Next, to ensure that differences in HCP and PNC scan coverage (HCP: 9 nodes lacked coverage in one or more subjects; PNC: 18 nodes lacked coverage in one or more subjects; Supplementary Figs. 2 and 3) did not affect results, the analysis was repeated for the HCP data after excluding the 9 additional nodes that lacked coverage in the PNC data set. Results were largely unchanged (Supplementary Table 5, "250 node"). To investigate the potential effects of parcellation resolution on main results, a 600-node parcellation was applied to the PNC data and CPM was repeated; again, results were largely unchanged (Supplementary Table 5, "600 node"), suggesting that the resolution of the 268-node atlas does not limit model performance.

Given variations in condition duration, time courses from all conditions in a given data set were truncated to include the same number of frames as the shortest condition in that data set, connectivity matrices were recalculated using these truncated time courses, and CPM was repeated on these matrices. Overall, task-based models again outperformed rest-based models (Supplementary Table 6), suggesting that condition duration did not drive differences between task- and rest-based models' performance.

To ensure that results are robust to cross-validation approach, CPM was repeated using $k$-fold, rather than leave-one-out, cross-validation (Methods). Results were largely unchanged (Supplementary Table 7).

Finally, to empirically assess the potential effect of global signal regression on main results, CPM was repeated on connectivity matrices computed without global signal regression (Methods). While the pattern of results remained unchanged (i.e., task-based models outperformed rest-based models), model performance decreased substantially overall (Supplementary Table 8). This result, taken with decreased performance differences among task-based models, suggests that the global signal may represent an important confound[26] in the search for trait-relevant, task-specific changes in functional connectivity.

**Models generalize across conditions and data sets**. To test the generalizability of these models, two validations were performed: cross-condition and cross-data set. In all cases, edges that passed thresholding ($P < 0.01$) in the training data were selected from the test data and used to predict gF (Methods). WRAT and PVRT scores were also used to test whether cross-condition patterns generalize, and whether models built on one intelligence-related measure generalize to another. In all tested combinations, task-based models proved robust enough to yield significant cognitive predictions (all $P < 0.01$) when applied to data acquired during different conditions (Fig. 1i), or to data acquired during similar conditions in a different data set (Fig. 1j).

In the cross-condition analysis, the models derived from the best-performing condition shared across data sets (WM) were applied to data from the worst-performing condition (rest; "WM to rest"), and vice versa ("rest to WM"; Methods). Interestingly, rest to WM better predicted gF than did WM to rest (HCP: WM to rest, $r_s^2 = 1.4\%$ ($P < 0.01$); rest to WM, $r_s^2 = 3.5\%$ ($P < 2e−5$); PNC: WM to rest, $r_s^2 = 2.8\%$ ($P < 6e−5$); rest to WM, $r_s^2 = 4.1\%$ ($P < 1e−6$); Fig. 1i). The same pattern held for cross-condition WRAT and PVRT predictions (Fig. 1i). In fact, in all but 2 out of 12 tested cases, the rest-based models performed better when applied to the WM data than when applied to the rest data for which they were built (see Fig. 1d, h), suggesting that brain state, as determined by task, affects model performance more than does precise edge selection.

In the cross-data set analysis, models derived from one data set were applied to data from the corresponding condition in the other data set (e.g., HCP WM models were applied to PNC WM data ("HCP to PNC") and vice versa ("PNC to HCP")), with analyses limited to the three conditions and nodes that were shared across data sets (Methods). Given the substantial differences among HCP and PNC tasks and subject populations (Discussion), it is noteworthy that cross-data set prediction was successful (HCP to PNC: WM, $r_s^2 = 6.4\%$ ($P < 9e−10$); emotion, $r_s^2 = 3.9\%$ ($P < 2e−6$); and rest, $r_s^2 = 3.9\%$ ($P < 3e−6$); PNC to HCP: WM, $r_s^2 = 2.7\%$ ($P < 2e−4$); emotion, $r_s^2 = 3.6\%$ ($P < 2e−5$); and rest, $r_s^2 = 1.1\%$ ($P < 0.02$); Fig. 1j). Task-based models built using Pmat scores in the HCP data yielded significant (all $P < 2e−5$) predictions of PVRT and WRAT scores in the PNC data ($r_s^2 = 3.4 – 12.6\%$; Fig. 1j), and task-based models built using PVRT and WRAT scores in the PNC data yielded significant (all $P < 0.002$) predictions of Pmat in the HCP data ($r_s^2 = 2.1−3.7\%$; Fig. 1j); rest-based models only yielded significant (all $P < 0.02$) predictions when HCP-based models were used to predict PVRT and WRAT in the PNC data (and not when PVRT-based and WRAT-based models were used to predict Pmat in the HCP data; Fig. 1j). In all, these results suggest that task-based CPMs of intelligence generalize across data sets and cognitive measures.

**Model edges are spatially distributed and overlapping**. We next sought to understand the relative contributions of different brain regions to these models. The edges included in each model are widely distributed throughout the brain (Supplementary Fig. 4). Nevertheless, considering that there are over 30,000 edges eligible for selection, overlap among models (computed for both the CN and AN; Methods) is substantial, with the greatest within-data set overlap between models derived from WM and emotion tasks (PNC, 11.02–11.78%; HCP, 6.93–8.71%; Fig. 2a); cross-data set overlap is also greater for models built from task than rest data (task: 3.43–4.31%, rest: 1.18–1.37%; Fig. 2a).

Given the models' sprawling distributions, measures drawn from network analysis are useful to summarize network structure. Here, we used node degree, computed as the number of edges incident to that node, to identify hubs, or nodes that are involved

in many connections in a given model (Methods). While the widespread distribution of these predictive networks is again underscored by this analysis, we highlight several trends.

First, within each data set, patterns of node degree were relatively stable across 1000 iterations of a split-half prediction analysis (Supplementary Fig. 1), and qualitatively similar across conditions (Fig. 2c). This similarity was confirmed by the high correlation of node degree across conditions and data sets, particularly between task-based models (within data sets: $r_s = 0.294$–$0.389$ (all $P < 2e{-}6$); between data sets: $r_s = 0.166$–$0.279$ (all $P < 0.01$); Fig. 2b). In addition, degree maps demonstrate substantial bilateral symmetry, as indicated by the correlation of node degree across hemispheres ($r_s = 0.17$–$0.62$); task-based models demonstrate greater symmetry than rest-based models (Methods; Supplementary Table 9).

Second, the mean degree of better performing models is right-shifted, and hub regions are similar across conditions (higher degree is represented with darker colors in Fig. 2c). This is consistent with the approximately linear, positive relationship between model performance and number of selected edges (Supplementary Fig. 5), and suggests that these additional edges in high-performing models belong to a coherent network or set of networks that is differentially perturbed by tasks. Taken with the similarity of node degree patterns across conditions and data sets, these findings further suggest that CPM is identifying a core gF-related network differentially perturbed by each condition.

Interestingly, the correlation between CN and AN degree vectors for a given condition was comparable to or in many cases substantially greater than cross-condition CN–CN and AN–AN degree correlations (HCP: WM, $r_s = 0.657$ ($P < 3e{-}33$); emotion, $r_s = 0.561$ ($P < 8e{-}23$); rest, $r_s = 0.292$ ($P < 2e{-}6$); PNC: WM, $r_s = 0.579$ ($P < 9e{-}24$); emotion, $r_s = 0.453$ ($P < 5e{-}14$); rest, $r_s = 0.422$ ($P < 4e{-}12$)), suggesting that the CN and AN in fact represent a single network in which overall patterns of functional connectivity are linearly related to gF.

An additional way to explore model structure is to identify canonical brain networks that contribute disproportionately to predictive models. To do so, we used ten functional networks derived from the same healthy subjects used to define the 268-node atlas (Methods; Supplementary Fig. 6) and, for each pair of networks, computed the fraction of edges in the given model that belong to that pair, normalized by the fraction of total edges belonging to that pair. Edge counts are thus scaled such that a value of 1 indicates proportionate contribution of that network pair to the model (Fig. 2d).

The results again reflect the distributed nature of these models. Additionally, in most models, visual (networks 5–7) and motor (4) regions are overrepresented. HCP networks tend to demonstrate greater involvement of frontal regions (1–2) than do corresponding PNC networks, particularly in the emotion task and rest CNs. Finally, this analysis sheds further light on the relationships between the CN and AN for a given condition. We

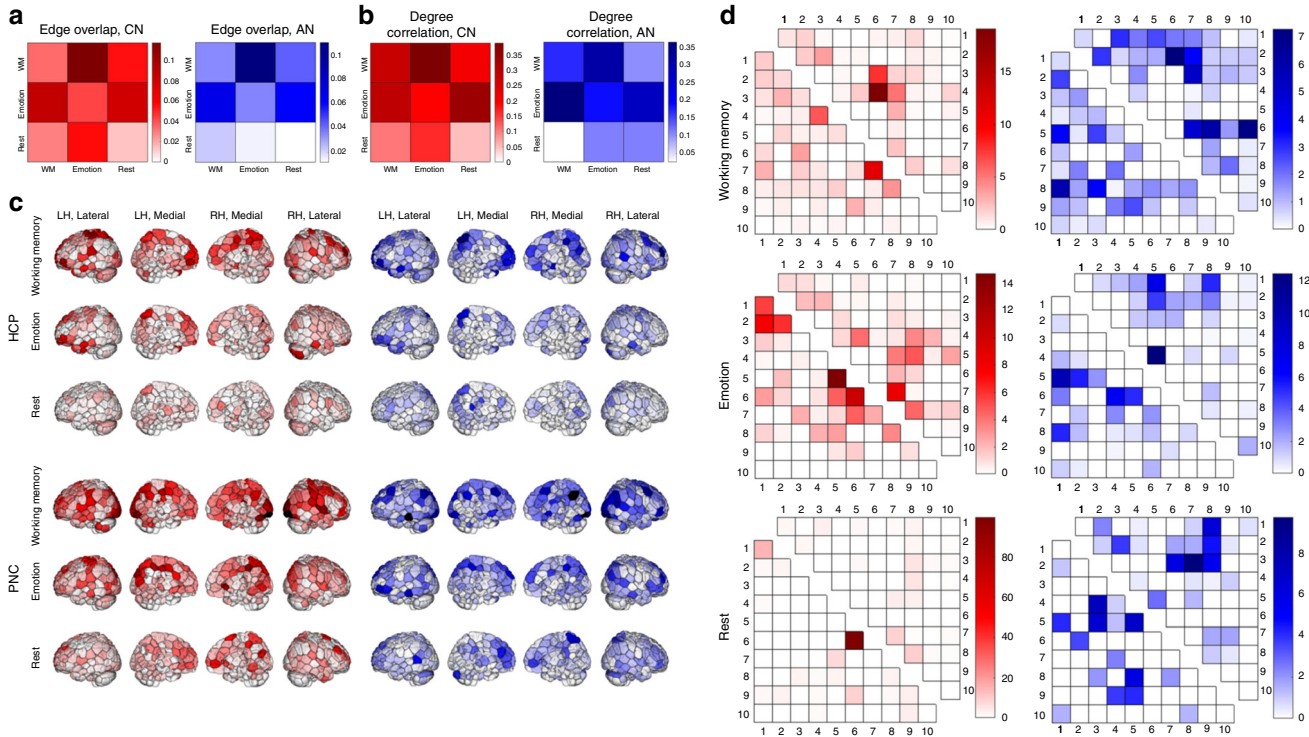

**Fig. 2** Model connections are widely distributed throughout the brain and demonstrate substantial overlap between models. **a** Edge overlap (number of shared edges normalized by the total number of unique edges in the models) between each pair of models within (off-diagonal) and between (main diagonal) data sets. In these and all subsequent matrix visualizations, HCP data are presented in the bottom triangle and PNC data are presented in the upper triangle. CN, correlated network; AN, anti-correlated network. **b** Spearman's correlation of node degree between each pair of models both within (off-diagonal) and between (main diagonal) data sets. **c** Visualization of node degree for each model in the HCP (top three rows) and PNC (bottom three rows) data. CN degree is displayed in warm colors; AN degree is displayed in cool colors; darker color indicates higher degree. LH, left hemisphere; RH, right hemisphere. **d** Canonical networks that contribute disproportionately (i.e., value > 1; see main text) to each model. As in **a** and **b**, HCP models are represented in the lower triangles and PNC models in the upper triangles. Each number corresponds to one canonical network (Methods): 1 = medial frontal, 2 = frontoparietal, 3 = default mode, 4 = motor cortex, 5 = visual A, 6 = visual B, 7 = visual association, 8 = salience, 9 = subcortical, 10 = cerebellum. Models in **a**–**c** were generated with an edge-selection threshold of $P < 0.01$; models in **d** were generated with an edge-selection threshold of $P < 0.001$ for improved visualization and interpretability

predicted, given the high degree correlation between CN/AN pairs, that the same networks would be overrepresented in these pairs, but that their patterns of connectivity with other networks would differ. This was indeed the case. For example, medial frontal (1) nodes are overrepresented in the HCP emotion task CN and AN, but these nodes tend to be connected with medial frontal (1) and frontoparietal (2) regions in the CN, and with visual (5) and salience network (8) regions in the AN. Similarly, visual regions (6–7) are overrepresented in both the PNC WM CN and AN, but their connections with motor cortex (4) and the default mode network (3) are overrepresented in the CN, while their connections with frontoparietal regions (2), salience network regions (8), and the cerebellum (10) are overrepresented in the AN.

**The best task for gF prediction varies by sex.** We next investigated whether patterns of model performance are consistent across a population or differ by group. Perhaps the most salient demographic feature, and one that has received much attention in the FC literature[10,27,28], is sex; to test the effect of sex on model performance, we divided each sample by sex (HCP: 241 males, 274 females; PNC: 251 males, 320 females), and performed CPM separately for males and females. In both the HCP and PNC data sets, we found marked and consistent sex differences in model performance. Interestingly, among the shared tasks across data sets, males and females demonstrated opposite patterns of model

performance: emotion task-based models outperformed WM task-based models in females (PNC: emotion $r_s^2 = 11.8\%$ ($P < 3e-10$), WM $r_s^2 = 6.3\%$ ($P < 6e-6$), Steiger's $z = 2.163$ ($P < 0.02$); HCP: emotion $r_s^2 = 5.9\%$ ($P < 5e-5$), WM $r_s^2 = 0.5\%$ ($P > 0.05$), Steiger's $z = 2.357$ ($P < 0.01$); edge-selection threshold of $P < 0.01$ (see Supplementary Note 1); Fig. 3a, c), while WM task-based models outperformed emotion task-based models in males (PNC: WM $r_s^2 = 9.7\%$ ($P < 6e-7$), emotion $r_s^2 = 4.0\%$ ($P < 0.005$), Steiger's $z = 2.349$ ($P < 0.01$); HCP: WM $r_s^2 = 20.3\%$ ($P < 3e-13$), emotion $r_s^2 = 7.3\%$ ($P < 3e-5$), Steiger's $z = 3.016$ ($P < 0.005$); edge-selection threshold of $P < 0.01$ (see Fig. 3a, c, Supplementary Note 1)). There were no significant sex differences in mean frame-to-frame displacement (all $P > 0.05$, Bonferroni corrected), as determined by two-tailed $t$-test, suggesting that systematic differences in head motion do not explain these findings.

As in the whole-sample analyses (see Model edges are spatially distributed and overlapping), visualization of node degree demonstrates that these networks are broadly distributed, with substantial similarities among models within sex group, and better performing models again demonstrating right-shifted mean degree in all but one case (PNC female emotion vs. WM task-based ANs; Fig. 3b, d).

## Discussion

In two large, independent data sets, we have demonstrated that CPMs built from task-based fMRI data better predict individual

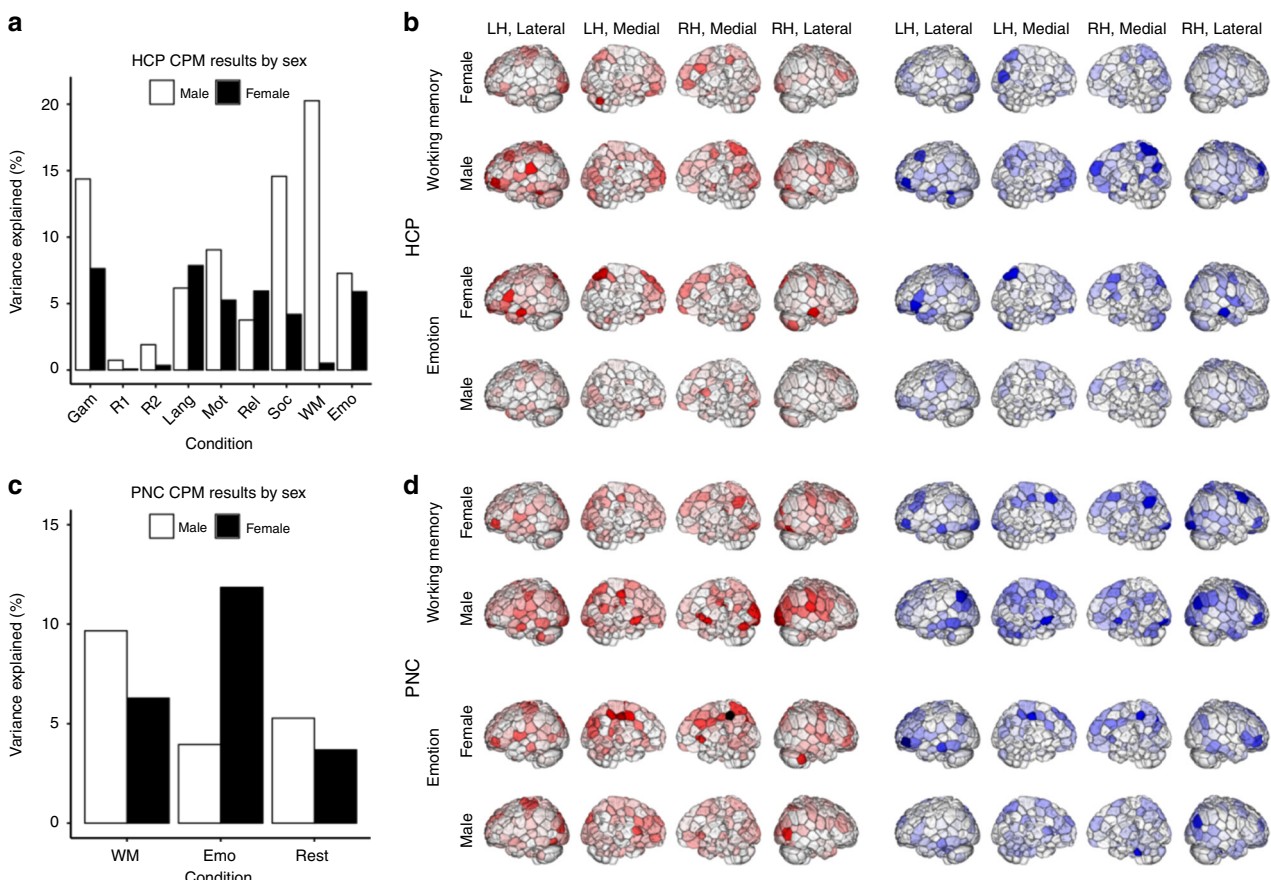

**Fig. 3** Distinct networks, best perturbed by different tasks, underlie fluid intelligence in males and females. **a** Results from the CPM pipeline run separately for males ($n = 241$) and females ($n = 274$) on data from each of the 9 HCP conditions (edge-selection threshold of $P < 0.01$). Abbreviations as in Fig. 1. **b** Visualization of node degree for male and female HCP models. Abbreviations and conventions as in Fig. 2c. **c** Results from the CPM pipeline run separately for males ($n = 251$) and females ($n = 320$) on data from each of the 3 PNC conditions (edge-selection threshold of $P < 0.01$). Abbreviations as in Fig. 1. **d** Visualization of node degree for male and female PNC models. Abbreviations and conventions as in Fig. 2c

traits (here, intelligence-related measures) than those built from resting-state fMRI data. Insofar as tasks modulate brain state, this finding suggests that brain state manipulations can yield important information about individual differences in brain functional organization and cognition. That is, state manipulations reveal trait differences. Further, some task-based models performed better than others, and the task that yielded the best predictions of gF varied by sex. Among the tasks shared across data sets, CPMs built from data acquired during a WM task consistently outperformed those built from data acquired during an emotion task in males, while the opposite was true in females. In all, this suggests that task-induced changes in functional connectivity can be task-specific and, taken with group features such as sex, informative when developing predictive models of individual traits: by using the right task in a sex-specific sample, the best model in these analyses explained over 20% of the variance in gF, as compared to <6% of variance explained by rest-based models built using the whole sample.

These findings lend support to the theory that though task-evoked changes in functional connectivity may comprise small perturbations of a stable, intrinsic network architecture robust to brain state, these changes are functionally relevant[29–31]. In fact, these results push this idea one step further, suggesting that task-induced changes in functional connectivity not only subserve performance of the task at hand (as evidenced, for example, by better prediction of task performance from task than rest data[17]), but also amplify individual differences in neural circuitry underlying related traits, differences that may not be detectable at rest. We highlight that this conclusion is revealed by but not specific to the CPM pipeline; rather, it transcends methodological approach to suggest that carefully selected cognitive tasks improve resolution of such differences and correspondingly permit more complete and robust characterization of brain–behavior relationships.

These findings involved no correction for task-related activation[32], suggesting that such activations do not hinder—and perhaps even improve—prediction in the CPM framework. While interpretation of such an analysis in task data alone may be limited by the inability to separate individual differences in intrinsic and task-induced connectivity, the use here of data acquired during both task execution and rest ensures that any improvement in prediction with task- relative to rest-based models derives from task-induced changes in functional connectivity and individual differences in these modulations. Moreover, these results are in spite of substantially longer resting-state scans than task scans in the HCP data set, which would be expected to increase reliability of functional connectivity estimates[33,34] in the resting state and thus improve rest-based model performance[35].

That task-based models outperform rest-based models likely reflects, in large part, the unconstrained nature of the resting state[12]. Functional connectivity variability is greater during rest than tasks, a finding that has been suggested to demonstrate increased mind wandering at rest[36]; recent experiences and brain states significantly alter patterns of resting-state functional connectivity[31,37,38]; and in contrast to the task-relevant[39], distinct patterns of connectivity identified during task states, resting-state connectivity patterns are better characterized by the joint expression of many states[40]. In short, rest is messy, and patterns of functional connectivity derived from it likely reflect many influences—arousal, attention, high-level processes associated with conscious thought—that remain difficult to measure. Conversely, tasks offer a controlled manipulation of brain state that taps into relevant circuitry[39]; any individual differences in this circuitry will be amplified, facilitating the prediction of related traits.

The utility of specific tasks for trait prediction is highlighted by the finding of sex differences in model performance: emotion task-based models outperform WM task-based models for females, while the opposite is true for males. This finding extends previous work on sex differences in functional connectivity[10,27,28], and may reflect sex differences in gF-related neural circuitry, task-related neural circuitry, or both. While sex differences in task-related circuitry may exist, they likely do not fully explain sex differences in model performance, as the spatial distributions of gF-related circuits were found to vary by sex (Fig. 3b, d). Further, the incorporation of sex into the models (Methods) failed to improve their performance (Supplementary Table 10). Together, these findings suggest that there are fundamental sex differences in the spatial distribution and modulation of gF-related networks (as suggested previously[41]) that cannot be captured in a single linear model. Further exploration of the relative contributions of sex differences in gF- and task-related circuitry to differences in model performance represents an important area for future investigation.

While sex differences in functional connectivity that correspond to sex differences in cognition have previously been reported, including in these data[27], we note that the current findings extend this work by leveraging these differences for trait prediction in unseen subjects, and by showing, given the improvements in prediction when the sexes are treated separately, that males and females are recruiting distinct networks to represent the same construct. This suggests that prediction performance may improve when models are built separately for males and females, and highlights the need to build the right model for each particular group. While group is here defined by sex, there are likely other relevant features that define groups. This may have important implications when CPM is applied clinically, as different patient populations may require distinct models to predict symptoms and behaviors. These differences may themselves, in turn, help categorize patients into relevant subgroups, consistent with the recent demonstration that patterns of functional connectivity can be used to identify and assign patients to treatment-relevant subtypes of depression[42].

A predictive model is of course most useful if it performs well and consistently across samples. Overestimation of model performance is common; this is in large part due to overfitting and a failure to maintain independence of training and test data[5,43]. By keeping training and test data separate at every step of the analysis, we ensured that every prediction was a true test of the models' ability to generalize to unseen subjects.

However, given that connectivity analyses usually involve many more edges than subjects, overfitting is difficult to eliminate completely[43], and likely contributes to the differences between rest-based CPM performance in these and previous[22] analyses that used a smaller data set[44]. The described validations—both cross-condition and cross-data set—were employed to further protect against overfitting, providing more strenuous tests of model generalizability.

Cross-condition prediction results demonstrate that models generalize across brain states, and that brain state may be even more important than edge selection; that is, even edges that do not survive thresholding in the WM data are, when summed, more systematically related to gF than are the most gF-relevant edges at rest. This explanation is consistent with the finding that edge-selection threshold does not affect model performance (Supplementary Table 1).

Differences between the HCP and PNC data make cross-data set prediction a harder problem, still, as the sample age ranges and experimental designs differed (even the tasks here treated as shared across data sets are in fact somewhat different; for example, the HCP WM task used images of faces, places, tools,

and body parts, while the PNC WM task used images of geometric figures). As expected, model performance decreased in these analyses, particularly when using PNC models to predict gF from HCP data (likely due to differences in data quality and developmental confounds that render the PNC models less robust), but prediction remained significant overall, even when using task-based models constructed from one intelligence-related measure to predict another. Validation results thus suggest that the models—particularly those built from task data—are robust to the particulars of the task, intelligence measure, and data set.

In addition to its prediction performance, CPM is compelling in its simplicity, which facilitates model summarization and interpretation[5]. Here, we have explored the spatial distribution of the described models and make several key observations. First, consistent with past reports[22,45–48], gF-related networks are spatially distributed across the brain, making whole-brain, data-driven approaches particularly well suited to their elaboration and application. Second, despite this distribution and the many relevant connections, overlap among models—both within and between data sets—is substantial, and greater for task- than rest-based models. Model overlap likely reflects the combined influences of a core set of trait-relevant edges, state-induced network reconfiguration[30], and methodological limitations; the relative contributions of each have yet to be determined, and present an opportunity for future investigation. Third, CN–AN pairs likely represent a single network in which a particular edge strength profile predicts gF. Finally, the anatomical distribution of the models, while complex, is largely consistent with existing accounts of the neuroanatomy underlying intelligence: visuomotor regions are overrepresented in gF-related circuitry, consistent with these regions' prominent inclusion in network models of intelligence (e.g.,[49]), the demonstrated relationship between motor skills and gF[48], and the relevance of visual cortex connectivity to the rule application phase of fluid intelligence testing[50]; and the greater contribution of frontal regions to the HCP networks than to the PNC networks agrees with the reported role of frontal cortex in intelligence[51–53], and with the emergence of this role in adulthood[52].

We have demonstrated that functional connectivity changes in trait-relevant ways with changing brain state, here induced by distinct tasks. There are many ways to characterize brain state, and many temporal resolutions at which it can be measured. As tools to study dynamic functional connectivity continue to advance[54], future investigations may seek to characterize brain state continuously, or at finer temporal scales to account for moment-to-moment fluctuations and their relationship to task design. Such work will be complemented by the use of continuous performance tasks that yield uninterrupted task-induced brain states. While the performance of the models described here suggests that such characterization of brain state may not be necessary for successful predictive modeling, it would likely facilitate interpretation of the models and advance our understanding of the relationships among brain state, functional organization, cognition, and behavior.

Further, while the application of a 600-node parcellation to the PNC data suggested that parcellation resolution does not substantially affect model performance, other improvements in registration and parcellation—such as the use of individualized parcellations[55], and of areal features for alignment and parcellation[56]—may improve the delineation of functionally homogeneous regions, as recently demonstrated in the HCP data[56], and correspondingly improve CPM performance. The impact of alignment and parcellation approaches on predictive model performance thus remains an important area for future investigation.

In addition, we have shown that our models generalize across two very different data sets, but it is likely that our results would improve with further characterization of relevant state changes and group features—that is, using the right tasks for the right subjects. Every data set is, by necessity, limited to a particular set of tasks and subject population. Development of models in additional, diverse data sets (e.g., with different tasks, older subjects, or patient populations) will thus be useful to further refine the models and explore the trends described here.

In summary, we have shown that brain state can be manipulated via cognitive tasks to perturb functional connections in the brain, better revealing brain-behavior relationships and allowing improved prediction of individual traits. The task that yields the best connectome-based predictive models varies by sex, suggesting that both subject group and brain state perturbations should be considered in functional connectivity and predictive modeling analyses. Previous work has modeled clinical symptoms in the same manner[17], suggesting the broad relevance of this work and the exciting possibility that task-based manipulations of brain state could assist in characterizing neural underpinnings of behavior and clinical symptoms across a wide range of psychiatric and neurological disorders.

## Methods

**Data sets**. Two data sets were used in the primary analyses described here: the Human Connectome Project (HCP) 900 Subjects release, which was the most recent HCP data release available at the time that this work began, and the Philadelphia Neurodevelopmental Cohort (PNC) first study release, which was the only data release available at the time that this work began. These data sets are described below.

**HCP participants**. From this data set, we used behavioral and functional imaging data[19]. We restricted our analyses to those subjects who participated in all nine fMRI conditions (seven task, two rest), whose mean frame-to-frame displacement was less than 0.1 mm and whose maximum frame-to-frame displacement was less than 0.15 mm (see HCP imaging parameters and preprocessing), and for whom gF measures were available ($n = 515$; 241 males; ages 22–36 + ). This conservative threshold for exclusion due to motion was used to mitigate the substantial effects of motion on functional connectivity; following this exclusion, there was no significant correlation between motion and gF for most conditions (all $P > 0.05$, Bonferroni corrected) except the social task, right-left (RL) phase encoding run ($r_s = -0.16$ ($P = 0.00017$)), the relational task, left-right (LR) phase encoding run ($r_s = -0.15$ ($P = 0.0008$)), and the emotion task, RL phase encoding run ($r_s = -0.14$ ($P = 0.0017$)).

**HCP imaging parameters and preprocessing**. Details of imaging parameters[19,57,58] and preprocessing[58,59] have been published elsewhere. In brief, all fMRI data were acquired on a 3 T Siemens Skyra using a slice-accelerated, multiband, gradient-echo, echo planar imaging (EPI) sequence (TR = 720 ms, TE = 33.1 ms, flip angle = 52°, resolution = 2.0 mm$^3$, multiband factor = 8). Images acquired for each subject include a structural scan and eighteen fMRI scans (working memory (WM) task, incentive processing (gambling) task, motor task, language processing task, social cognition task, relational processing task, emotion processing task, and two resting-state scans; two runs per condition (one LR phase encoding run and one RL phase encoding run))[58,60] split between two sessions. Each condition was a different length (WM, 5:01; gambling, 3:12; motor, 3:34; language, 3:57; social, 3:27; relational, 2:56; emotion, 2:16; rest, 14:33; see Effects of condition duration for further investigation of the potential implications of variable scan duration). The scanning protocol (as well as procedures for obtaining informed consent from all participants) was approved by the Institutional Review Board at Washington University in St. Louis. Use of HCP data for these analyses was deemed exempt from IRB review by the Yale Human Investigation Committee. The HCP minimal preprocessing pipeline was used on these data[59], which includes artifact removal, motion correction, and registration to standard space. All subsequent preprocessing was performed in BioImage Suite[61] and included standard preprocessing procedures[22], including removal of motion-related components of the signal; regression of mean time courses in white matter, cerebrospinal fluid, and gray matter; removal of the linear trend; and low-pass filtering. Mean frame-to-frame displacement was averaged for the LR and RL runs, yielding nine motion values per subject; these were used for subject exclusion and motion analyses. All subsequent analyses and visualizations were performed in BioImage Suite[61], Matlab (Mathworks), and R[62,63].

**PNC participants**. From this data set, we used behavioral, structural imaging, and functional imaging data[20]. We restricted our analyses to those subjects who participated in all three fMRI runs (two task, one rest), on whom registration was successful (nine subjects were excluded for failed registrations), whose mean frame-to-frame displacement was less than 0.1 mm and whose maximum frame-to-frame displacement was less than 0.15 mm (as for the HCP data set, and with the same motivation), and for whom fluid intelligence (gF) measures were available ($n = 571$; 251 male, ages 8–21). Following exclusion for motion, there was no significant correlation between motion and gF for any condition (all $P > 0.05$, Bonferroni corrected).

**PNC imaging parameters and preprocessing**. Details of the imaging protocol have been published elsewhere[64]. In brief, all fMRI data were acquired on a 3 T Siemens TIM Trio using a multi-slice, gradient-echo EPI sequence (TR = 3000 ms, TE = 32 ms, flip angle = 90°, resolution = 3 mm$^3$). During each imaging session, subjects completed a structural scan and three fMRI scans (WM task, emotion identification task, and resting-state scan). As in the HCP data set, each condition was a different length (WM, 11:39; emotion, 10:36; rest, 6:18). The potential implications of this were explored (see Effects of condition duration). All study procedures, including protocols for obtaining informed consent from all participants, were approved by the Institutional Review Boards at the University of Pennsylvania and the Children's Hospital of Philadelphia. As for the HCP analyses, use of the PNC data for these analyses was deemed exempt from IRB review by the Yale Human Investigation Committee. Standard preprocessing procedures were applied to these data. Structural scans were skull stripped using an optimized version of the FMRIB's Software Library (FSL)[65] pipeline[66]. Slice time and motion correction were performed in SPM8[67]. The remainder of image preprocessing was performed in BioImage Suite[61] and included linear and nonlinear registration to the MNI template; regression of mean time courses in white matter, cerebrospinal fluid, and gray matter; and low-pass filtering. All subsequent analyses and visualizations were performed in BioImage Suite[61], Matlab (Mathworks), and R[62,63].

**Functional parcellation and network definition**. The Shen 268-node atlas derived from an independent data set using a group-wise spectral clustering algorithm[21] was applied[22] to both the preprocessed HCP and PNC data. After parcellating the data into 268, functionally coherent nodes, the mean time courses of each node pair were correlated and correlation coefficients were Fisher transformed, generating nine 268 × 268 connectivity matrices per HCP subject, and three 268 × 268 connectivity matrices per PNC subject (one per fMRI run; matrices were generated for both the LR and RL phase encoding runs in the HCP data, and these matrices were averaged for each condition). Of note, in a subset of subjects in each data set, some of these nodes lacked sufficient coverage (the scan volume was too restricted); we adopted the conservative approach of excluding these nodes in all subjects. In the HCP data, 9 nodes lacked sufficient coverage, and were dropped from all further HCP analyses. Nine additional nodes lacked sufficient coverage in the PNC data (for a total of 18 nodes with incomplete coverage in the PNC data); these 18 nodes were dropped from all further PNC and cross-data set analyses. These nodes were primarily in subcortical regions (Supplementary Figs. 2 and 3).

The same spectral clustering algorithm was used to assign these 268 nodes to 8 networks[21,22], and the subcortical-cerebellar network was split into networks 8–10[34] (Supplementary Fig. 6); these networks are named based on their approximate correspondence to previously defined resting-state networks, and are numbered for convenience according to the following scheme: 1. Medial frontal, 2. Frontoparietal, 3. Default mode, 4. Motor, 5. Visual A, 6. Visual B, 7. Visual association, 8. Salience, 9. Subcortical, 10. Cerebellum. Numbers of nodes in these networks are presented in Supplementary Table 11.

**Cognitive prediction**. In both data sets, fluid intelligence was quantified using matrix reasoning tests. In the HCP data set, a 24-item version of the Penn Progressive Matrices test was used; this test is an abbreviated form of Raven's standard progressive matrices[68]. In the PNC data set, 24- and 18-item versions of the Penn Matrix Reasoning Test were used[69,70]. Integer scores indicate number of correct responses (HCP: PMAT24_A_CR, range = 5–24, mean = 17.53, s.d. = 4.45, median = 19; PNC: PMAT_CR (phv00194834.v1.p1.c1), range = 0–23, mean = 12.27, s.d. = 4.04, median = 12). Percent, rather than number, correct (PMAT_PC (phv00194837.v1.p1.c1)) was also used in a follow-up analysis of the PNC data (see Effects of gF measurement technique).

Connectome-based predictive modeling (CPM)[18,22] was first carried out for each data set separately. Briefly, iterative, leave-one-subject-out cross-validation (LOOCV) was used to predict gF in the left-out subject. The first step of this pipeline is edge selection. The strength of each edge in $n-1$ subjects was related to gF in those subjects using Pearson correlation; the edges with the strongest positive correlations with gF were assigned to the "correlated network" (CN), while those with the strongest negative correlations with gF were assigned to the "anti-correlated network" (AN). This step requires that the significance of edges' correlations with gF be thresholded, and given the inevitably arbitrary nature of this threshold, various thresholding methods (based on number of edges (i.e., sparsity) or correlation $P$ value) and levels (1%, 2.5%, 5%, 10%, $P < 0.01$, $P < 0.005$, $P < 0.001$) were tested to ensure that variable network size across

conditions and subject groups did not account for differences in model performance.

Next, "network strength" was calculated for the CN and AN for each subject in the training group by summing the weights of all CN and AN edges in each subject's connectivity matrix, yielding two such summary statistics for each subject. The difference between these network strengths was calculated to yield a "combined network" strength measure:

$$\text{CN strength}_s = \sum_{i,j} \mathbf{c}_{i,j} \mathbf{m}_{i,j}^+ \tag{1}$$

$$\text{AN strength}_s = \sum_{i,j} \mathbf{c}_{i,j} \mathbf{m}_{i,j}^- \tag{2}$$

$$\text{Combined net strength}_s = \text{CN strength}_s - \text{AN strength}_s \tag{3}$$

where $\mathbf{c}$ is the connectivity matrix for subject $s$, and $\mathbf{m}^+$ and $\mathbf{m}^-$ are binary matrices indexing the edges $(i,j)$ that survived thresholding for the CN and AN, respectively.

Linear regression was then used to evaluate the relationship between network strength and gF in the same $n-1$ subjects, yielding a first-degree polynomial that best fit network strength to gF in a least squares sense; three such models were built, one each for the CN, AN, and combined network.

In the final step, CN, AN, and combined network strengths were calculated for the excluded subject, and were submitted to the corresponding models to generate three gF estimates for that subject. This process was repeated iteratively, with each subject excluded once, and the entire pipeline was repeated for every condition (task and rest) in each data set (Fig. 1a–h).

Model performance was quantified as the Spearman's correlation between predicted and true gF ($r_s$). An alternative way of expressing this quantity is in terms of explained variance ($r_s^2$). For clarity and improved interpretability, we often refer to the latter, and note that conditions that yielded negative correlations between predicted and true gF (i.e., conditions that could not be used to build a successful linear model of gF) are assumed to explain none of the variance in gF scores, and $r_s^2$ is correspondingly set to zero.

Because the iterations of CPM are not independent, the significance of model predictions in the main gF CPM analyses was assessed via 1000 iterations of permutation testing, with $P$ equal to the fraction of iterations on which the correlation between predicted and true (permuted) gF was greater than the correlation between predicted and true (unpermuted) gF. To account for family structure in the HCP and PNC data sets, gF permutations were only permitted among siblings of the same type (i.e., non-twin, dizygotic twin, or monozygotic twin in the case of HCP data, and non-twin or twin in the case of PNC data), and among families with identical structure[23,24]. Of note, one individual in the PNC sample was missing family information; because it seems unlikely that this information would have been collected for one family member and not for another, this individual was assumed to not have siblings in the sample and coded accordingly. Resulting $P$ values were corrected for multiple comparisons using the false discovery rate[25]; both $P$ and $q$ values are presented in the text (see Results, State manipulations improve trait predictions) to facilitate comparison with previous work, in which only rest-based model performance was assessed[22,34]. The significance of the overall difference between task- and rest-based models' performance was assessed via Mann-Whitney $U$ test performed on rest- and task-based model performance measures (i.e., $r_s^2$), pooled across both data sets. Except as otherwise noted, this and all subsequent $P$ values are presented in their uncorrected form, given the post-hoc nature of these analyses. The stability of main gF CPM results and model anatomy (see Analysis of anatomical distribution of model edges) was evaluated via split-half prediction (i.e., for each data set, the data were randomly divided in half, models were trained on one half and tested on the other, and this procedure was repeated 1000 times). Results are presented in Supplementary Fig. 1.

CPM was repeated for two alternative intelligence-related measures (WRAT (range = 70–145, mean = 103.22, s.d. = 15.78, median = 102) and PVRT (range = 4–15, mean = 11.68, s.d. = 2.52, median = 12); Table 1) in the PNC data. These WRAT and PVRT analyses were considered post-hoc, and the corresponding $P$ values of these and all subsequent analyses, except as otherwise noted, were thus computed parametrically.

**Validation of the models**. As discussed in the main text, two validations of these models were performed: cross-condition and cross-data set. In all cases, models were generated using an edge-selection threshold of $P < 0.01$. We chose to use a less conservative threshold to minimize the potential effects of overfitting and noise introduced by trait-irrelevant differences (e.g., in task design).

In the cross-condition analyses, models generated from the HCP WM condition (training data) were applied to the HCP rest matrices (test data), and vice versa (Fig. 1i). The same analyses were performed using the PNC WM and rest data (Fig. 1i). Models were generated using the training data from $n-1$ subjects, and these models (CN, AN, and combined network) were applied to the test data for the left-out subject, as described previously (see Cognitive prediction). For example, in the WM to rest analysis, WM task-based connectivity matrices were

used for edge selection and model building in $n-1$ subjects, and these edges were then selected from the rest-based connectivity matrix in the left-out subject. Their weights were summed to yield CN, AN, and combined network strengths for the left-out subject, which were submitted to the corresponding models to generate predictions of the given subject's trait measure.

In the cross-data set analyses, only data from rest, the WM task, and the emotion processing task were used. Models generated from each of these conditions in the HCP data (training data) were applied to the corresponding condition from the PNC data (test data), and vice versa (Fig. 1j). Given the missing nodes in the two data sets, only the 250 nodes common to both data sets were used in all cross-data set analyses. Training data networks were generated by taking the intersection of the binary matrices $\mathbf{m}^+$ and $\mathbf{m}^-$ from all iterations of CPM for the given condition. These networks were applied to the test data, and the strengths of included edges were summed for each subject, yielding CN, AN, and combined network strength scores for each subject. Model performance was then quantified as the Spearman's correlation between network strength and gF which, given the independence of the training and test data, is equivalent to prediction. As noted previously, models for which combined network strength was negatively correlated with gF are assumed to explain none of the variance in gF, and $r_s^2$ is set to zero. Thus, these analyses yielded three measures of model performance (for the CN, AN, and combined network) for each of the four cross-condition analyses (HCP WM to rest, HCP rest to WM, PNC WM to rest, PNC rest to WM) and six cross-data set analyses (HCP to PNC WM, PNC to HCP WM, HCP to PNC emotion, PNC to HCP emotion, HCP to PNC rest, and PNC to HCP rest). This analysis was repeated using WRAT and PVRT scores in the PNC data (i.e., cross-condition analyses as described above, and cross-data set analyses in which PNC models based on edge correlation with the given measure were used to predict gF in the corresponding condition of HCP data, and in which HCP gF models were used to predict the given measure in the corresponding condition of PNC data).

**Analysis of sex differences in model performance**. To investigate a potential sex effect in model performance, we repeated the analysis described above (Cognitive prediction) with the sample split by sex. Differences between model pairs' (i.e., WM task- and emotion task-based models for a given sex group) performance were assessed for significance using Steiger's $z$[71] and corresponding one-tailed $P$ values, as implemented by Lee and Preacher[72]. Given the finding of substantial sex differences in model performance (Fig. 3), we next sought to improve model performance by explicitly incorporating sex into the CPM pipeline. To do so, we first adapted the edge-selection step to use partial correlation, rather than correlation, with sex as a covariate (i.e., to control for any effect of sex on the relationship between edge strength and gF, and thus select edges that are correlated with gF in a sex-independent manner in the HCP data, and with sex and age as covariates in the PNC data. This manipulation did not affect model performance (Supplementary Table 10), and is not discussed further here. Next, we adapted the model-building step to incorporate sex as a second, binary predictor in a multilinear regression. Similarly, this manipulation did not improve model performance (Supplementary Table 10), and is not discussed further here. Finally, we compared mean frame-to-frame displacement in males and females using two-sample $t$-tests assuming equal variances, given results of two-sample $F$ tests for equal variance for each condition.

**Effects of head motion**. Given the finding that correlations between gF and mean frame-to-frame displacement did not meet significance in 18 out of 21 runs of the HCP and PNC data sets, but that model predictions were frequently correlated with mean frame-to-frame displacement, we sought to better understand the potential effects of subject motion on model performance. As in the analysis of sex differences, we explicitly incorporated mean frame-to-frame displacement for each subject into the CPM pipeline, both at the edge-selection step (again via partial correlation) and at the model-building step (again via multilinear regression). Results are presented in Supplementary Table 4.

**Effects of gF measurement technique**. Given that two versions of the Penn Matrix Reasoning Test were used to assess gF in the PNC (one with 18 items and one with 24 items), we repeated the analysis described above (Cognitive prediction) with the sample split by test version. In addition, as in the sex differences and motion analyses, we sought to improve model performance by incorporating test version into the CPM pipeline via partial correlation (now with version as a covariate) and via multilinear regression (now with the inclusion of a binary version predictor in the models). We also repeated the CPM analysis using percent, rather than raw number, correct to train and test the models. Finally, we repeated the analysis after excluding subjects whose Pmat scores were valid but who experienced a problem that may have affected performance (version code "V2," $n = 9$) and those whose Pmat scores were not valid (version code "N," $n = 2$). All results are presented in Supplementary Table 2.

**Effects of parcellation resolution and scan coverage**. It was hypothesized that increasing the resolution of the parcellation may correspondingly improve model

performance by allowing more subtle individual differences in patterns of FC to emerge. This prediction was tested in the PNC data set by applying a 600-node parcellation[73] to the data and repeating the CPM analyses as previously described (Cognitive prediction). Results are presented in Supplementary Table 5.

Further, given the difference in spatial coverage of the acquisitions in the HCP and PNC data (the latter only included coverage of 250 out of 268 nodes; see Functional parcellation and network definition), it was verified that this difference did not account for any observed differences in model performance by limiting the HCP matrices to the same 250 nodes included in the PNC matrices and repeating the CPM analyses described above (Cognitive prediction). Results are presented in Supplementary Table 5.

**Effects of HCP reconstruction method and quality issues**. Given the development of an improved image reconstruction algorithm during HCP Phase II scanning, we repeated the CPM analyses described above (Cognitive prediction) including only data reconstructed using the r227 algorithm ($n = 402$). Next, as previously described, we sought to improve model performance by incorporating the version of the reconstruction algorithm into the CPM pipeline via partial correlation (now with algorithm version as a covariate) and via multilinear regression (now with the inclusion of a binary algorithm version predictor in the models). Results are presented in Supplementary Table 3. Finally, given known quality control (QC) issues, we excluded HCP subjects with QC Issues B, C, and D (i.e., subjects with focal segmentation and surface errors, subjects for whom data were collected during a time of coil temporal instability, and subjects who demonstrated artifact in minimally preprocessed resting-state fMRI data, as described on the HCP wiki), as well as subjects missing gradient-recalled echo field maps (again, as described on the HCP wiki). CPM was re-run on this sample subset ($n = 475$); results are presented in Supplementary Table 3.

**Effects of condition duration**. Because every condition in each data set was a different length, we sought to explore the potential effects of condition duration on corresponding model performance. To do so, we correlated duration and model performance; this correlation was found to be negative in the HCP data and positive in the PNC data, an effect apparently driven by rest runs that were longer than task runs in the HCP data, and task runs that were longer than rest runs in the PNC data. That task-based models outperformed rest-based models in both cases suggests that this effect is not driven by condition duration, but to further explore the potentially confounding effects of condition duration, we truncated each time course to include the same number of frames as the shortest condition in that data set (HCP: 176 frames; PNC: 124 frames) and recalculated connectivity matrices using these truncated time courses. These matrices were submitted to the CPM pipeline, and results are presented in Supplementary Table 6.

**Effects of cross-validation method**. To explore the bias-variance trade-off inherent in selecting the cross-validation method[44], we repeated the analysis described above (Cognitive prediction) using a $k$-fold, rather than a leave-one-out, cross-validation approach to model training and testing, with $k = 10$. That is, for each data set, the sample was divided into 10, approximately equally sized groups; on each fold, the model was trained on 9 groups and tested on the excluded 10th group. This process was repeated iteratively, with each group excluded once. Results are presented in Supplementary Table 7.

**Effects of global signal regression**. In light of the controversy surrounding the use of global signal regression (GSR)[74], we repeated the main CPM analyses using matrices that were computed without GSR for a subset of the HCP sample used in the main analyses ($n = 514$). Results are presented in Supplementary Table 8.

**Analysis of anatomical distribution of model edges**. We next sought to understand how different brain regions may contribute to these networks, and to evaluate the overlap among models built from different conditions and data sets. Except as otherwise noted, we used the intersection networks (see Validation of the models) generated with edge-selection thresholds of $P < 0.01$. As in the cross-condition and cross-data set analyses, we chose to use a less conservative threshold to minimize the potential effects of overfitting and noise.

First, we calculated model edge overlap between different conditions within each data set (e.g., between PNC WM and PNC emotion task-based models) and between the same condition in different data sets (i.e., HCP WM and PNC WM task-based models, HCP emotion and PNC emotion task-based models, and HCP rest- and PNC rest-based models). CNs and ANs were treated separately. For each model pair, we counted the number of shared edges, and normalized this value by the total number of unique edges in the models (Fig. 2a). Because some of the models were substantially larger than others (e.g., Supplementary Fig. 5), we repeated this analysis using the models generated with sparsity thresholds. The same patterns held (e.g., WM and emotion task-based models shared the greatest fraction of edges within and between data sets, while rest-based models generally demonstrated lower within- and between-data set overlap), though, as would be expected if models reflect fundamental, trait-relevant circuits, percent overlap increased with increasing network size. Finally, to ensure that the imposition of an edge-selection threshold did not itself impact overlap results, we correlated the

strength of each edge across all subjects with gF in those subjects, yielding a $1 \times e$ vector of correlation coefficients for all edges in each condition, where $e$ is the number of edges. We then correlated these vectors to quantify the similarity of gF-related edge distributions across conditions. Trends were comparable to those identified in the overlap analyses; these results are presented in Supplementary Fig. 7.

We next asked whether there is shared structure among models that cannot be captured at the edge level. One could imagine, for example, that a given node is connected with node $i$ in one model, and with node $i + 1$ in another; nodes $i$ and $i + 1$ may be neighboring nodes that belong to the same macroscale network, but these edges would of course not overlap, and thus this shared structure would be overlooked in the edge overlap analysis. To pursue this, we drew from network analysis approaches and calculated binary node degree[75] for each model:

$$D_i^+ = \sum_j \mathbf{m}_{i,j}^{+,\text{int}} \qquad (4)$$

$$D_i^- = \sum_j \mathbf{m}_{i,j}^{-,\text{int}} \qquad (5)$$

where $\mathbf{m}^{+,\text{int}}$ and $\mathbf{m}^{-,\text{int}}$ are again the intersection CN and AN, respectively; $D_i^+$ is the degree of the $i^{th}$ node in the CN, and similarly $D_i^-$ is the degree of the $i^{th}$ node in the AN. This yielded 12 node degree vectors: HCP WM CN, HCP WM AN, HCP emotion CN, HCP emotion AN, HCP rest CN, HCP rest AN, PNC WM CN, PNC WM AN, PNC emotion CN, PNC emotion AN, PNC rest CN, and PNC rest AN. The spatial distribution of node degree was visualized for each of the degree vectors by coloring nodes according to their degree (Figs. 2c, 3b, d), and the similarity of degree distributions between model pairs was quantified as the Spearman's correlation between the corresponding degree vectors (Fig. 2b). Bilateral symmetry of node degree was quantified by calculating the Spearman's correlation between the degree values of all left-hemisphere nodes and their right-hemisphere homologs, and vice versa. To assign homologs given the asymmetry of the parcellation, we calculated the distance between each node's centroid and the centroids of all nodes in the opposite hemisphere after reflecting them over the midline; the node in the reflected hemisphere that was closest to the given node was assigned as its homolog. Nodes that lacked coverage—along with their paired nodes —were excluded from further analyses. To assess the stability of node degree, we calculated degree vectors for each of the 1000 iterations of the split-half analysis and, for each condition, correlated degree vectors for every pair of iterations (Supplementary Fig. 1).

Finally, to explicitly explore the macroscale networks related to gF and perturbed by each task, we assigned each edge in $\mathbf{m}^{+,\text{int}}$ and $\mathbf{m}^{-,\text{int}}$ to a pair of canonical networks (i.e., edge $(i,j)$ would be assigned to the network that includes node $i$ and the network that includes node $j$; see Functional parcellation and network definition for an explanation of the canonical networks). Edge counts were normalized to account for network sizes as follows:

$$\text{Contribution}_{A,B}^+ = \frac{m_{A,B}^{+,\text{int}} / m_{\text{tot}}^{+,\text{int}}}{E_{A,B} / E_{\text{tot}}} \qquad (6)$$

$$\text{Contribution}_{A,B}^- = \frac{m_{A,B}^{-,\text{int}} / m_{\text{tot}}^{-,\text{int}}}{E_{A,B} / E_{\text{tot}}} \qquad (7)$$

where $\text{Contribution}_{A,B}^+$ and $\text{Contribution}_{A,B}^-$ represent the relative contributions of connections between canonical networks $A$ and $B$ to the intersection CN and AN, respectively; $m_{A,B}^{+,\text{int}}$ and $m_{A,B}^{-,\text{int}}$ are the numbers of edges between A and B in the intersection CN and AN, respectively; $m_{\text{tot}}^{+,\text{int}}$ and $m_{\text{tot}}^{-,\text{int}}$ are the total numbers of edges in the intersection CN and AN, respectively; $E_{A,B}$ is the number of edges between $A$ and $B$ in the whole brain; and $E_{\text{tot}}$ is the total number of edges in the whole brain. Because these analyses do not evaluate network overlap and sparse networks facilitate interpretation of results, these steps were performed on networks generated with an edge-selection threshold of $P < 0.001$. Assignments are visualized in a $10 \times 10$ matrix. These are diagonally symmetric matrices, and therefore we only display the bottom triangle of each HCP matrix and the upper triangle of each PNC matrix (Fig. 2d).

**Code availability**. Matlab scripts to run the main CPM analyses can be found at (https://www.nitrc.org/projects/bioimagesuite/). BioImage Suite tools used for analysis and visualization can be accessed at (http://bisweb.yale.edu). Matlab scripts written to perform additional post-hoc analyses are available from the authors upon request.

**Data availability**. The HCP data that support the findings of this study are publicly available on the ConnectomeDB database (https://db.humanconnectome.org). The PNC data that support the findings of this study are publicly available on the database of Genotypes and Phenotypes (dbGaP, accession code phs000607.v1.p1); a data access request must be approved to protect the confidentiality of participants.

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

## Acknowledgements

Data were provided in part by the Human Connectome Project, WU-Minn Consortium (Principal Investigators: David Van Essen and Kamil Ugurbil; 1U54MH091657) funded by the 16 NIH Institutes and Centers that support the NIH Blueprint for Neuroscience Research; and by the McDonnell Center for Systems Neuroscience at Washington University. The remainder of the data used in this study were provided by the Philadelphia Neurodevelopmental Cohort (Principal Investigators: Hakon Hakonarson and Raquel Gur; phs000607.v1.p1). Support for the collection of the data sets was provided by grant RC2MH089983 awarded to Raquel Gur and RC2MH089924 awarded to Hakon Hakonarson. All subjects were recruited through the Center for Applied Genomics at The Children's Hospital in Philadelphia. This work was also supported by NIH Medical Scientist Training Program Training Grant T32GM007205 (A.S.G.). Many thanks to Corey Horien, Xilin Shen, Monica Rosenberg, Evelyn Lake, Daniel Barson, Xenophon Papademetris, Mehraveh Salehi, and Stephanie Noble for their assistance with and feedback on this project.

## Author contributions

A.S.G., S.G., D.S., and R.T.C. conceptualized the study. A.S.G. and S.G. performed the analyses with support from D.S. D.S. and R.T.C. provided guidance on result interpretation and follow-up analysis design. A.S.G. wrote the manuscript, with contributions from D.S. and R.T.C. and comments from S.G

## Additional information

**Competing interests:** The authors declare no competing interests.

