## [Peer Review File · Nature Communications]

Reviewers' comments:

Reviewer #1 (Remarks to the Author):

This is an interesting and potentially very important study that uses "connectome-based predictive modeling" (CPM) to examine predictability of fluid intelligence (gF) in two large and independent cohorts (HCP and PNC). The main assertions are that (i) task-fMRI assessments of functional connectivity (FC) performs significantly better than resting-state fMRI in predicting traits related to intelligence; (ii) specific tasks differ in their trait predictiveness; and (iii) there are sex differences in which tasks perform best. More generally, the authors argue that their findings motivate 'a paradigm shift' from resting-state to task-based FC analyses. This is indeed a credible suggestion, but it makes it all the more important for the current study to be on as solid a footing as possible.

While many of the results are associated with very low p-values (high significance) and are thus likely to be robust, this is not uniformly the case. Moreover, there are several methodological concerns that are important to address.

Major concerns.

1) HCP family structure. Based on the methods described in the current ms and also in the more general approach described in Nature Protocols (Shen et al, 2017), it appears that the authors have not taken into account the family structure of the HCP cohort (twins and non-twin siblings) in their statistical analyses. Failure to account for family structure can lead to false positives and inflated estimates of statistical significance Winkler et al. (Neuroimage, 2015). Taking family structure into account is unlikely to have a dramatic effect in this particular study, but in this reviewer's assessment it is essential in order for the statistics to be on a solid footing. Moreover, since the authors are promoting the CPM approach as a useful tool for other investigators to use, it is all the more incumbent upon them to make the CPM tools more robust for the community at large. The authors might find this URL helpful: https://fsl.fmrib.ox.ac.uk/fsl/fslwiki/PALM/ExchangeabilityBlocks#EBs_for_data_of_the_Human_Connectome_Project

2) In Fig. 1, the difference in trait predictability for tasks versus rest is reasonably clear and impressive for most tasks in both HCP and PNC cohorts. The text (p. 6) states that the differences are significant for all but the HCP relational task. However, the significance is borderline ($p = .04$) for the HCP social vs R1. Besides the aforementioned concern about accounting for family structure, another methodological issue is whether these p values were corrected for multiple comparisons. Item 14 in the Additional Material indicates that this was indeed done; it should also be mentioned in the Methods (I didn't find it).

3) The contrast between task and rest is all the more impressive in view of the fact that for HCP, the individual tasks were much shorter in duration than the rest scans. The authors later make a point of this (Discussion, p. 14) but should give specifics in the Methods and also indicate whether the apparent differences between tasks is correlated with individual task duration.

Much rides on the claims that there are convincing differences across tasks in trait predictiveness. The statement at the bottom of p. 6 that "certain tasks yielding better predictions than others" may be technically correct, but it appears to rest on one leg: HCP gambling is better than relational tasks. While this one contrast appears to be highly significant, it is critical to know how general this is. At this place in the ms, the conclusion seems overstated.

4) The sex differences illustrated in Fig. 3 are fascinating and appropriately come later in the

presentation. However, the scatterplots in Fig. 1b-d and f-h offer an intriguing opportunity to plot M vs F subjects in different colors (e.g., red, green, and yellow where they overlap). This might prove to be quite informative without taking up additional figure space; discussion of the sex differences could still be postponed to later in the text.

5) p. 7. Another possible control relates to whether the results depend specifically on use of regressing “global” signals (gray and white matter plus CSF; cf. Methods, p. 19). This step forces a zero-mean correlation and is controversial in the field. In the Shen (2017) paper the authors note in passing that there are alternative approaches such as partial correlation and ICA-based spatial denoising (e.g., Smith et al, NN 2015). Ideally, the authors would be able to run partial correlation for at least some of their tests to see if the results differ markedly. At a minimum they should justify their choices and note these potential confounds.

6) Edge overlap. Fig. 2 provides a data-rich comparison of edge overlap and node degree within and across models and cohorts. The edge overlap (Fig. 2a) is generally quite modest (~12% is the highest and most are much lower). Of course, there is inherently a lot of noise in these datasets, but it leaves this reviewer uneasy that the spatial patterns of connectivity appear to be pretty low in reproducibility. The edge overlap is a binary measure that entails thresholding and discounting of edge weights. The alternative of estimating overlap using weighted measures without thresholding might in principle have greater sensitivity and should at least be considered and mentioned.

7) Node degree overlap. A striking observation (p. 9 and Fig. 2 b, c) is that CN and AN degree vectors are highly correlated. A potential confound is that CNR/SNR is regionally variable and dependent on the head coil and pulse sequence. Higher CNR regions should presumably tend to have higher degree in general, related to data acquisition characteristics, not just neurobiological factors. This issue warrants careful consideration.

8) Sex differences. p. 12, Fig. 3, and Table 2. Some of the sex differences for different tasks are impressively large, highly significant, and consistent to some degree across HCP and PNC cohorts. However, the M vs F difference for WM is very large for HCP but quite modest for PNC (and not stated as to whether the latter difference is significant). Similarly, the M vs F difference for emotion is large for the PNC cohort (but significance not stated) whereas there is hardly any difference for the HCP cohort.

On p. 12, para. 3, the authors report ‘preliminary evidence for a sex-by-age interaction’ However, the data in Table 2 appear distinctly underwhelming and not even a consistent trend for females, and the claims about ‘outperforming’ noted in the table legend should either be tempered or buttressed by p-values. This paragraph and table should perhaps be dropped or relegated to SI.

9) Somewhere in the discussion the authors should discuss whether some of the lower variance explained for rest be related to differences in attention or arousal states, as the rest state may associated with reduced arousal, drowsiness, and even overt sleep for part of the scan in some/many subjects.

10) Limitations (pp. 17-18), parcellation (p. 20) and parcellation resolution (pp. 23-24). The authors use a 268-node whole-brain parcellation (‘functional atlas’) from Shen et al. (2013), which is based on a volume-based alignment across individuals. For the PNC data, they compare results for a 600-node parcellation (Craddock et al., 2012) and find similar results. Their implicit conclusion that the choice of parcellation and of alignment method doesn’t matter much is overstated. Methods that achieve more accurate alignment of functional subdivisions (e.g., the areal-feature-based alignment for cortical structures in the publicly released HCP data) and the associated multimodal parcellation (Glasser et al, 2016) might well yield better predictive power. This issue

warrants mention so that readers are aware that alignment quality and parcellation accuracy are important issues that may impact and eventually contribute to improved CPM performance.

11) Image reconstruction confound. p. 23. Another HCP-specific confound that needs to be addressed (but seems to have been overlooked) is that early in the HCP the fMRI image reconstruction was changed to an improved method. See <https://wiki.humanconnectome.org/display/PublicData/Ramifications+of+Image+Reconstruction+Version+Differences>. It should be straightforward to regress out this confound.

Minor comments.

p. 3, end of para. 1. "...ideal..." is not an 'ideal' word. "Attractive" would be better.

Fig. 1a and 3a. Please rotate the text to oblique or vertical and increase font size along x axis for legibility. On Fig. 1e, simply increase font size.

Fig. 1i: The unfilled bars (WM to rest) are barely visible. Use a fill shading that increases visibility while preserving contrast with Rest to WM bars.

p. 7, para. 2, pp. 20-21, and Table S2: Clarify whether the two versions of the PMAT test refer to just the PNC cohort. Was one of the PNC versions identical to the HCP PMAT24 version?

Fig. 2a, b. Can't read units on scales. Please enlarge font.

p. 11 para. 2. The FC measure is not directed, so please use 'patterns of connectivity with' rather than '....to' here and elsewhere.

Methods. Please state key parameters for HCP and PNC scans: Spatial resolution, multiband factor, TR, scan durations, to spare readers from needing to look these up.

Reviewer #2 (Remarks to the Author):

"Task-induced brain state manipulation improves prediction of individual traits" is an extension of the authors' previously published study on resting state fMRI fingerprints. Here they examine whether models built from task fMRI data perform better than those built from resting-state fMRI data. This a nice direction, but neither the results nor approach is novel, as several previous studies have examined this question (Cole et al and others). Further, this study has several other major limitations: it is not theoretically motivated, tests no neural hypotheses, and the findings are incremental and inconsistent with respect to a previous publication from the authors (Finn et al., 2015 NN). Surprisingly, findings from that study are not discussed, although the authors employed almost exactly the same pipeline and investigated the same question "do individual connectivity profiles predict fluid intelligence". Overall, the study is weak and findings generally uninterpretable. Other concerns include:

(1) There is no theoretical justification for using gambling, motor, social, and emotion tasks to predict fluid intelligence. Gambling task activation turned out to a strong predictor of gF. I wonder what the interpretation and theory here is.

(2) As in Finn et al., the results demonstrate that connectivity profiles can predict fluid intelligence. However, if we compare Finn et al. Fig. 5a with current Fig. 1d, it appears that models built from the previous small sample actually outperform those from the larger sample used here. This suggests that

predictive values of connectivity-features (or the effectiveness of the current pipeline) decreases with increasing sample size? Moreover, in Finn et al study, just using features from fronto-parietal networks achieved the same performance as using features from the whole brain (Fig. 5a vs. 5c). Although the authors did not test if this is the case in the current study, it appears that visuomotor regions are more highly predictive of fluid intelligence in the current study (Fig. 2d). Does this fit any cognitive theory or models of fluid intelligence? I doubt it.

(3) Use of a developmental PNC cohort with a large age range is problematic.

(4) The authors employed a univariate approach to select features using connectivity features that show a significant correlation with fluid intelligence. This is highly circular and problematic, and a likely reason for the inflated findings in Finn et al. This is now quite problematic as the current results suggest a failure to replicate original Finn et al. findings.

Reviewer #3 (Remarks to the Author):

Review of: Task-induced brain state manipulation improves prediction of individual traits

I read this article with great interest not least because I agree strongly with the central premise, which is that dynamic/task active connectivity should provide a better predictor of cognitive ability than resting state connectivity. This notion makes a great deal of sense, given that we see patterns of large-scale network activity/coherences during the performance of g-loaded tasks. The analyses and results have a number of important strengths, however, there are also a number of key points that I believe if addressed would greatly strengthen the article.

1) The authors state that "By considering task-induced brain state and sex, the best-performing model explains over 20% of the variance in fluid intelligence scores, as compared to less than 5% of variance explained by rest-based models built using the whole sample."

This is an intriguing result, particularly given the WM/emotional task dissociation. The replication of the task*gender interaction across the two studies is a strength. My main concern though is that the difference in variance explained when accounting for gender could relate to differences in cohort size. By definition, there must be fewer subjects in the trained datasets for the gender analyses, i.e., because the cohort has been split into two groups. Is it not the case that they will be more prone to overfit, and therefore, could give the illusion of explaining more variance? To be truly convincing, the trained model would really need to be validated by application to a further dataset. I also was curious why only gender was examined as opposed to other factors (handedness, age, education level, etc?)

2) The main finding is that "brain state can be manipulated to better reveal brain-behavior relationships, and that identifying and inducing the right brain state in a given group can improve trait prediction".

This seems sensible to me, g-loaded tasks involve activity/coherence across certain networks, it makes sense to examine those networks when they are expressing those active/synchronised states. The fact that "Results generalize across conditions and two large, independent datasets" is also a strength - the importance of reproducibility is finally coming to the fore in the imaging field at the moment, which is a good thing. A concern though, is how well balanced this comparison actually is? Specifically, were the rest and task acquisitions the same duration? I.e., did they have the same acquisition parameters, and were there the same type and number of EPI scans for task and rest? This is an important point to consider. Longer acquisition will allow for a more robust estimate of correlation strength between each pair of nodes. This results in a connectivity matrix that has greater signal vs noise. One should ensure that the rest vs task difference is not simply a consequence of

differences in the reliability of these estimates.

3) Relatedly to the above two points, when stating how much variance is actually explained, I would have been more convinced if the headline value was for a cross validation. by this, I mean where the model is trained on HCP data for one task, and then validated with more HCP data for that same task. I don't question that there is a connectivity-g relationship, but without such a step, the headline estimates of actual variance explained are hard to interpret (I note, that this is a criticism that can be levelled at a number of other high impact papers, and the authors at least apply a cross validation across studies, albeit with somewhat different tasks/acquisitions).

4) The comparison of rest vs. task seems well balanced and sensible (assuming the above condition regarding number of TRs is met). A question is whether the authors also looked at measures taken of dynamic connectivity relative to steady state (PPI for example), or were the average taken across both rest and task for the task acquisitions? In a way it is critical to use the simple correlational as opposed to PPI approach, as this is more balanced across task and rest conditions, making them more comparable. Nonetheless, an obvious prediction of the authors hypothesis is that the PPI measures (increased connectivity during task vs rest) should in fact give the best possible explanation of g.

5) I was somewhat confused by the description of supplementary analyses with more robust compensation for movement artefacts. Does this mean that movement still had a component in the connectivity-g relationship for the primary analyses as reported in the main text? if so, then really these supplementary analyses should be presented as the main analyses, and headline statistics altered accordingly. If not, and the minimal data cleaning pipeline was sufficient, then are these not redundant?

6) The observation that models trained on rest data provide stronger prediction of 'g' when applied to task data seems to be a strength - although I was not clear on how generally / consistently this was the case - it would be good to clarify this in the text. Also, to clarify that the task and rest acquisitions actually have the same amount of data as outlined above.

7) Repeating analyses with different parcellation resolutions is a strength of the paper.

8) It is surprising that leave-one-out approach was applied in such a large dataset - the gold standard would be to use cross validation with train and test sub-groups.

9) I was unclear whether the number of edges that were feature selected in the male and female populations was balanced? Obviously any differences in this would lead to a problem, whereby more edges gives the model more degrees of freedom for fitting (and over-fitting) the data.

In summary this is an intriguing and potentially important paper with great potential. I hope that the above suggestions are helpful in strengthening it further.

We thank the reviewers for their thoughtful comments. These comments motivated additional analyses and revisions that we feel substantially strengthened the manuscript, and we highlight these changes below. Corresponding changes in the manuscript are indicated using blue text.

Reviewer 1:

This is an interesting and potentially very important study that uses “connectome-based predictive modeling” (CPM) to examine predictability of fluid intelligence (gF) in two large and independent cohorts (HCP and PNC). The main assertions are that (i) task-fMRI assessments of functional connectivity (FC) performs significantly better than resting-state fMRI in predicting traits related to intelligence; (ii) specific tasks differ in their trait predictiveness; and (iii) there are sex differences in which tasks perform best. More generally, the authors argue that their findings motivate ‘a paradigm shift’ from resting-state to task-based FC analyses. This is indeed a credible suggestion, but it makes it all the more important for the current study to be on as solid a footing as possible.

While many of the results are associated with very low p-values (high significance) and are thus likely to be robust, this is not uniformly the case. Moreover, there are several methodological concerns that are important to address.

Major concerns.

1) HCP family structure. Based on the methods described in the current ms and also in the more general approach described in Nature Protocols (Shen et al, 2017), it appears that the authors have not taken into account the family structure of the HCP cohort (twins and non-twin siblings) in their statistical analyses. Failure to account for family structure can lead to false positives and inflated estimates of statistical significance Winkler et al. (Neuroimage, 2015). Taking family structure into account is unlikely to have a dramatic effect in this particular study, but in this reviewer’s assessment it is essential in order for the statistics to be on a solid footing. Moreover, since the authors are promoting the CPM approach as a useful tool for other investigators to use, it is all the more incumbent upon them to make the CPM tools more robust for the community at large. The authors might find this URL helpful:

<https://fsl.fmrib.ox.ac.uk/fsl/fslwiki/PALM/ExchangeabilityBlocks#EBs> for data of the Human Connectome Project

We agree that performing significance tests with clarity and rigor is of utmost importance, and thank the reviewer for pointing out the issue of restrictions on exchangeability. To address this issue, we have implemented the block permutation pipeline of Winkler and colleagues^{1,2} for main analyses in both the HCP and PNC data sets. In the latter, one subject out of the 571 was missing family information; because it seems unlikely that this information would have been collected from one family member but not another, we assumed that this subject did not have siblings in the cohort and coded this subject as such. A description of this approach has been added to Methods (Cognitive prediction, p24). Revised *P* values based on these permutation tests, as well as FDR-corrected *q* values, are reported in the text (Results, State manipulations improve trait predictions, p6).

2) In Fig. 1, the difference in trait predictability for tasks versus rest is reasonably clear and impressive for most tasks in both HCP and PNC cohorts. The text (p. 6) states that the differences are significant for all but the HCP relational task. However, the significance is borderline ($p = .04$) for the HCP social vs R1. Besides the aforementioned concern about accounting for family structure, another methodological issue is whether these p values were corrected for multiple comparisons. Item 14 in the Additional Material indicates that this was indeed done; it should also be mentioned in the Methods (I didn't find it).

The P values corresponding to results of main analyses in this manuscript (i.e., the CPM results depicted in Fig. 1a-h and described on page 6 of the main text) were FDR-corrected³; in the revised manuscript, in addition to stating overall q values at the beginning of the Results section entitled “State manipulations improve trait predictions” (p6), we have included both P and corrected q values throughout this section. We include both values for clarity and to facilitate comparison to our previous work^{4,5}, in which only rest data were used, and correction was thus not necessary (i.e., to avoid the illusion that significance of rest-based model prediction is different from that described in our previous work). Because of the sheer number of follow-up and confound analyses, all other analyses were considered to be post-hoc, and P values are thus presented without correction, except as otherwise indicated. Methods (Cognitive prediction, p24) and Results (p5) have been revised to clarify this point.

3) The contrast between task and rest is all the more impressive in view of the fact that for HCP, the individual tasks were much shorter in duration than the rest scans. The authors later make a point of this (Discussion, p. 14) but should give specifics in the Methods and also indicate whether the apparent differences between tasks is correlated with individual task duration.

We thank the reviewer for pointing out this oversight – we agree that the different durations of the conditions are important, and have included the length of each condition in the Methods sections entitled “Imaging parameters and preprocessing” (p21-22) for both the HCP and PNC data sets. To further explore the potential effects of condition duration on model performance, several additional analyses were performed. First, condition duration was correlated with model performance. For the HCP data, longer conditions yielded models with significantly worse performance ($r = -0.78$, $P < 0.05$), but this effect was dominated by the rest runs (correlation of duration and model performance for only task runs: $r = 0.23$, $P = 0.6$). For the PNC data, longer conditions yielded models with significantly better performance ($r = 0.999$, $P < 0.05$). That task-based models outperformed rest-based models both when task runs were longer than rest runs (PNC) and when task runs were shorter than rest runs (HCP) suggests that this effect is not driven by condition duration. However, to further explore this potential confound, time courses from all conditions within each data set were truncated to include the same number of frames as the shortest condition (HCP: 176 frames; PNC: 124 frames). Connectivity matrices were recalculated and submitted to the CPM pipeline; the pattern of results was largely unchanged (i.e., task-based models outperformed rest-based models, with the gambling task yielding the best-performing model in the HCP data, and the WM task yielding the best-performing model in the PNC data). These analyses are described briefly in Results (Investigation of potential confounds, p8) and more extensively in Methods (Effects of condition duration, p26-27), and corresponding results ($100r^2$) are presented in Supplementary Table 6 (reproduced below).

	Task	Truncated time courses
HCP	Gam	13.8
	R1	2.5
	R2	0
	Lang	8.2
	Mot	8.9
	Rel	4.5
	Soc	5.7
	WM	5.9
	Emo	9.8
PNC	WM	8.8
	Emo	7.7
	Rest	3.9

Much rides on the claims that there are convincing differences across tasks in trait predictiveness. The statement at the bottom of p. 6 that “certain tasks yielding better predictions than others” may be technically correct, but it appears to rest on one leg: HCP gambling is better than relational tasks. While this one contrast appears to be highly significant, it is critical to know how general this is. At this place in the ms, the conclusion seems overstated.

Our intention here was to demonstrate two points: task-based models outperformed rest-based models, and not all task-based models performed equally well. To avoid the many comparisons that would result from exhaustive comparison of each model pair, we chose to demonstrate the first point by comparing only the worst-performing task-based models and the best-performing rest-based model, and the second point by comparing only the best- and worst-performing task-based models in each data set. However, we appreciate that this approach is confusing, and may not adequately demonstrate these two points. In the revised manuscript, we have chosen to focus on the difference between task- and rest-based models’ performance, and to this end, have replaced these analyses with a Mann-Whitney U test comparing all task-based models to all rest-based models (pooled across both data sets). This clearly demonstrates that task-based models significantly outperformed rest-based models. These results are described on page 6 (“In both data sets, some tasks yielded better gF predictions than others, but in all cases, task-based models outperformed rest-based models (rank sum = 71, two-sided $P = 0.018$, Mann-Whitney U test).”), and this revised approach is described in Methods (Cognitive Prediction, p24).

4) The sex differences illustrated in Fig. 3 are fascinating and appropriately come later in the presentation. However, the scatterplots in Fig. 1b-d and f-h offer an intriguing opportunity to plot M vs F subjects in different colors (e.g., red, green, and yellow where they overlap). This might prove to be quite informative without taking up additional figure space; discussion of the sex differences could still be postponed to later in the text.

The colors of the points on the scatterplots in Figure 1 have been colored by sex (red for females, and blue for males). For clarity given the sheer number of subjects and the broad regions of overlap, we chose to use only two colors and increase the transparency of all points, but we

welcome additional suggestions about using a third color, should the reviewer feel that this would add helpful information to the figure.

5) p. 7. Another possible control relates to whether the results depend specifically on use of regressing “global” signals (gray and white matter plus CSF; cf. Methods, p. 19). This step forces a zero-mean correlation and is controversial in the field. In the Shen (2017) paper the authors note in passing that there are alternative approaches such as partial correlation and ICA-based spatial denoising (e.g., Smith et al, NN 2015). Ideally, the authors would be able to run partial correlation for at least some of their tests to see if the results differ markedly. At a minimum they should justify their choices and note these potential confounds.

Given evidence that global signal regression (GSR) effectively reduces motion artifact in fMRI⁶, we expected model performance to be improved by the inclusion of GSR in our preprocessing pipeline. Nevertheless, we appreciate the controversy surrounding the use of GSR; the most straightforward way to address the impact of this preprocessing choice on functional connectivity patterns and resulting CPM results is to repeat our main analyses using data that have not been subjected to GSR. We did so with the HCP data and found, as expected, that overall model performance decreased, but task-based models still outperformed rest-based models. This analysis is described in Methods (Effects of global signal regression, p27); results ($100r^2$) are summarized in Results (Investigation of potential confounds, p9) and presented in full in Supplementary Table 8, reproduced below:

		Feature-selection threshold		
Task		$P < 0.01$ ($n = 514$)	$P < 0.005$ ($n = 514$)	$P < 0.001$ ($n = 514$)
HCP	Gam	3.0	3.4	3.8
	R1	1.4	1.6	1.6
	R2	1.2	1.6	1.9
	Lang	4.4	4.8	5.5
	Mot	2.7	3.0	4.0
	Rel	3.1	3.6	4.2
	Soc	4.3	4.8	5.9
	WM	3.4	3.8	4.2
	Emo	3.5	2.9	2.8

We share this reviewer’s interest in partial correlation-based calculation of functional connectivity, not only because these approaches obviate the need for GSR, but also because they offer a data-driven method to more directly⁷ and sensitively⁸ measure connectivity between each node pair. To investigate the effects of connectivity estimation method on the main findings reported in this manuscript, we recalculated connectivity matrices using partial, rather than full, correlation. That is, we calculated the partial correlation between the mean time courses of every node pair, computed without GSR, such that each entry in the resulting connectivity matrix reflects the functional connectivity between the given node pair, controlling for all other nodes’ mean time courses.

While partial correlation-based functional connectivity measures have been compellingly validated⁸, we note that in many cases, including several of the conditions in these data sets, there are fewer observations (i.e., time points) than nodes. In such cases, regularization is required to estimate partial correlation matrices, which involves the selection of a tuning parameter that controls sparsity, or the degree of regularization. While this parameter is often selected empirically, the appropriate degree of regularization likely depends on the particularities of a given data set, and results have been found to vary substantially as this parameter is varied⁹. To our knowledge, there is as yet no standardized method for selection of this parameter, and a thorough exploration of this issue is beyond the scope of the current work; to avoid the potential pitfalls of arbitrary parameterization, we chose to calculate partial correlation-based functional connectivity only for those HCP conditions that yield full-rank time-by-node matrices (language task, motor task, social task, WM task, rest1, and rest2). Given the range in condition length and the sensitivity of partial correlation to data quantity, we truncated all time courses to the length of the shortest included condition (274 time points). We note that this number of time points is only slightly greater than the number of included nodes, which is likely to cause instabilities in results⁹. Further, to our knowledge, the use of partial correlation-based approaches to calculate task-based functional connectivity remains relatively unexplored. Given these concerns about the approach, we predicted that results would be noisy and unstable, and found this to be the case. These results ($100r^2$) are presented below:

		Feature-selection threshold		
		$P < 0.01$ ($n = 514$)	$P < 0.005$ ($n = 514$)	$P < 0.001$ ($n = 514$)
HCP	R1	0	9.3	7.3
	R2	0	0	2.0
	Lang	0	0	0
	Mot	3.1	0	3.3
	Soc	1.2	0	1.6
	WM	0	0	1.7

6) Edge overlap. Fig. 2 provides a data-rich comparison of edge overlap and node degree within and across models and cohorts. The edge overlap (Fig. 2a) is generally quite modest (~12% is the highest and most are much lower). Of course, there is inherently a lot of noise in these datasets, but it leaves this reviewer uneasy that the spatial patterns of connectivity appear to be pretty low in reproducibility. The edge overlap is a binary measure that entails thresholding and discounting of edge weights. The alternative of estimating overlap using weighted measures without thresholding might in principle have greater sensitivity and should at least be considered and mentioned.

To examine the potential impact of thresholding on quantification of model overlap, we performed an additional analysis in which we compared all edges' correlations with gF across conditions. That is, for each condition, we correlated each edge's strengths across all subjects with gF scores, yielding a single correlation coefficient for the given edge in that condition. We repeated this procedure for all edges and all conditions, yielding a $1 \times e$ vector of correlation coefficients for each condition, where e is the total number of edges. We then correlated these vectors to quantify the similarity of gF-related edge distributions across conditions. PNC

correlations ranged from $r = 0.37-0.51$, HCP from $r = 0.25-0.32$, and cross-data set (within condition) from $r = 0.12-0.19$, and the pattern of results was similar to that identified in the overlap analyses (i.e., task-based models demonstrated the greatest similarity both within and between data sets). These results are presented in Supplementary Figure 5 (reproduced below), and the approach is described in Methods (Analysis of anatomic distribution of model edges, p27-28). We note that, by both accounts, this overlap is substantial (particularly given that there are over 30,000 unique edges in the brain), and far greater than would be expected by chance, highlighting the existence of core gF-related circuitry that is differentially perturbed by different tasks.

Supplementary Figure 5. Similarity of edges' correlations with gF between conditions, demonstrating substantial overlap between models both within data sets (PNC data: upper triangle; HCP data: bottom triangle) and between data sets (main diagonal). Values indicate Spearman's correlation coefficients.

7) Node degree overlap. A striking observation (p. 9 and Fig. 2 b, c) is that CN and AN degree vectors are highly correlated. A potential confound is that CNR/SNR is regionally variable and dependent on the head coil and pulse sequence. Higher CNR regions should presumably tend to have higher degree in general, related to data acquisition characteristics, not just neurobiological factors. This issue warrants careful consideration.

This is an interesting and important potential issue that we have investigated in several ways. First, we leveraged previous analyses of temporal SNR (tSNR) in the HCP data¹⁰, and compared the resulting tSNR map (which was reportedly similar across the various tasks) to our maps of node degree. By visual inspection, the maps are quite distinct (e.g., inferior temporal (IT) cortex has low tSNR; in contrast, several hubs are found in IT cortex). Further, if SNR were driving node degree, degree correlation across conditions (Fig. 2b) would be expected to be quite high.

Comparison of HCP node degree (bottom; adapted from Fig. 2c) to a representative HCP tSNR map (top; adapted from Barch, D. M. *et al.* Function in the human connectome: task-fMRI and individual differences in behavior. *NeuroImage* **80**, 169–189 (2013)). [Editorial Note: This figure is reproduced with permissions from Elsevier (2013). All rights reserved.]

Next, as a proxy for SNR at the node level, we computed mean node reliability (i.e., the mean of reliability measured for every edge incident to the given node) in the HCP data⁴ and calculated the Spearman’s correlation between this measure and node degree for conditions shared across data sets (using a feature-selection threshold of $P < 0.01$ to facilitate comparison to Fig. 2b,c). The results are presented below, and demonstrate that degree was not significantly correlated with mean node reliability in any condition; insofar as reliability reflects SNR, this further suggests that node degree is not driven by SNR.

	Correlated network	Anti-correlated network
Working memory	$r = 0.106, P = 0.09$	$r = 0.035, P = 0.57$
Emotion	$r = 0.020, P = 0.74$	$r = 0.117, P = 0.06$
Rest 1	$r = 0.052, P = 0.40$	$r = -0.022, P = 0.72$

8) Sex differences. p. 12, Fig. 3, and Table 2. Some of the sex differences for different tasks are impressively large, highly significant, and consistent to some degree across HCP and PNC cohorts. However, the M vs F difference for WM is very large for HCP but quite modest for PNC (and not stated as to whether the latter difference is significant). Similarly, the M vs F difference for emotion is large for the PNC cohort (but significance not stated) whereas there is hardly any difference for the HCP cohort.

We thank the reviewer for pointing out this important point and apologize for any confusion caused by the presentation of these results. The primary goal of these analyses was to investigate,

among the conditions shared across data sets, whether the condition that yields the best gF predictions varies by sex. Overall, we found that CPMs performed better in HCP males than in HCP females ($\bar{r}(\text{males}) = 0.32$ versus $\bar{r}(\text{females}) = 0.22$). This main effect caused scaling differences that complicate the comparison of model performance for a given task between the sex groups (e.g., WM task-based model performance in males and females, as this reviewer suggests). Instead of pursuing this comparison, then, we chose to focus on the interaction between sex and condition (i.e., does the comparison between WM and emotion task-based models differ by sex?) to demonstrate that, for each group, a different task best perturbs circuitry relevant to the cognitive measure of interest.

On p. 12, para. 3, the authors report ‘preliminary evidence for a sex-by-age interaction’ However, the data in Table 2 appear distinctly underwhelming and not even a consistent trend for females, and the claims about ‘outperforming’ noted in the table legend should either be tempered or buttressed by p-values. This paragraph and table should perhaps be dropped or relegated to SI.

Given our shared concerns about the preliminary and inconsistent sex-by-age interaction in model performance, these results have been removed from the manuscript.

9) Somewhere in the discussion the authors should discuss whether some of the lower variance explained for rest be related to differences in attention or arousal states, as the rest state may associated with reduced arousal, drowsiness, and even overt sleep for part of the scan in some/many subjects.

We see differences in arousal and attention during rest to be important examples of the unconstrained nature of the resting “state,” and have added a brief exploration of this topic to the Discussion (p16):

That task-based models outperform rest-based models likely reflects, in large part, the unconstrained nature of the resting state¹¹. Functional connectivity variability is greater during rest than tasks, a finding that has been suggested to demonstrate increased mind wandering at rest¹²; recent experiences and brain states significantly alter patterns of resting-state functional connectivity^{13–15}; and in contrast to the task-relevant¹⁶, distinct patterns of connectivity identified during task states, resting-state connectivity patterns are better characterized by the joint expression of many states¹⁷. In short, rest is messy, and patterns of functional connectivity derived from it reflect many influences – arousal, attention, high-level processes associated with conscious thought – that remain difficult to measure. Conversely, tasks offer a controlled manipulation of brain state that taps into relevant circuitry¹⁶; any individual differences in this circuitry will be amplified, facilitating the prediction of related traits.

10) Limitations (pp. 17-18), parcellation (p. 20) and parcellation resolution (pp. 23-24). The authors use a 268-node whole-brain parcellation (‘functional atlas’) from Shen et al. (2013), which is based on a volume-based alignment across individuals. For the PNC data, they compare results for a 600-node parcellation (Craddock et al., 2012) and find similar

results. Their implicit conclusion that the choice of parcellation and of alignment method doesn't matter much is overstated. Methods that achieve more accurate alignment of functional subdivisions (e.g., the areal-feature-based alignment for cortical structures in the publicly released HCP data) and the associated multimodal parcellation (Glasser et al, 2016) might well yield better predictive power. This issue warrants mention so that readers are aware that alignment quality and parcellation accuracy are important issues that may impact and eventually contribute to improved CPM performance.

We agree that improved registration and parcellation may, in turn, improve model performance. We clarify in Results (Investigation of potential confounds, p8) and Methods (Effects of parcellation resolution and scan coverage, p26) that the comparison of results generated using the 268-node parcellation to results generated using the 600-node parcellation is solely an investigation of the potential effect of parcellation resolution on model performance. We have also added the following paragraph to the Discussion (p20):

Further, while the application of a 600-node parcellation to the PNC data suggested that parcellation resolution does not substantially affect model performance, other improvements in registration and parcellation – such as the use of individualized parcellations¹⁸, and of areal features for alignment and parcellation¹⁹ – may improve the delineation of functionally homogeneous regions, as recently demonstrated in the HCP data¹⁹, and correspondingly improve CPM performance. The impact of alignment and parcellation approaches on predictive model performance thus remains an important area for future investigation.

11) Image reconstruction confound. p. 23. Another HCP-specific confound that needs to be addressed (but seems to have been overlooked) is that early in the HCP the fMRI image reconstruction was changed to an improved method.

See <https://wiki.humanconnectome.org/display/PublicData/Ramifications+of+Image+Reconstruction+Version+Differences>. **It should be straightforward to regress out this confound.**

We thank the reviewer for pointing out this important potential confound that we indeed overlooked. To account for this difference in reconstruction algorithm, we performed three additional analyses (Results, Investigation of potential confounds, p7 and Methods, Effects of HCP reconstruction method and quality control issues, p26). First, we excluded those subjects for whom the r227 algorithm was not available, leaving 402 subjects on whom to perform the main CPM analyses. Next, we incorporated algorithm version into the feature-selection (via partial correlation) and model building (via multilinear regression) steps. These analyses are analogous to our exploration of the effects of head motion, sex, and gF measurement technique. CPM results were not substantially affected by these efforts to account for image reconstruction method; given the substantial number of subjects affected and the apparent lack of impact on our main results, we chose not to exclude subjects from analyses reported in the main text on the basis of reconstruction algorithm, but present these results in Supplementary Table 3. As an additional control, we re-ran the main CPM analyses after excluding all HCP subjects (40 in total) with QC Issues B, C, and D (for a description of these issues, see Methods, Effects of HCP reconstruction method and quality control issues, p26) or without gradient-recalled echo field

maps. Results ($100r^2$) were largely unchanged, and are presented in Supplementary Table 3, reproduced below:

	Task	No QC issues ($n = 475$)	r227 only ($n = 402$)	Partial correlation	Multilinear regression
HCP	Gam	14.0	10.2	11.3	12.6
	R1	2.6	1.9	3.2	2.8
	R2	0	0	0	0
	Lang	9.3	7.7	8.4	8.5
	Mot	8.2	6.5	9.0	8.6
	Rel	7.3	2.9	4.8	3.3
	Soc	6.2	6.0	6.1	6.2
	WM	11.1	11.6	10.6	10.5
	Emo	9.0	6.8	10.8	9.5

Minor comments.

p. 3, end of para. 1. “...ideal...” is not an ‘ideal’ word. “Attractive” would be better.

The word “ideal” has been replaced with the phrase “well suited.”

Fig. 1a and 3a. Please rotate the text to oblique or vertical and increase font size along x axis for legibility. On Fig. 1e, simply increase font size.

The x axis labels on Figs. 1a and 3a have been rotated, and the font enlarged. The x axis labels on Figs. 1e and 3c have been enlarged.

Fig. 1i: The unfilled bars (WM to rest) are barely visible. Use a fill shading that increases visibility while preserving contrast with Rest to WM bars.

The fill has been changed to light gray to ensure that the legend is still clear while enhancing visibility of the bars.

p. 7, para. 2, pp. 20-21, and Table S2: Clarify whether the two versions of the PMAT test refer to just the PNC cohort. Was one of the PNC versions identical to the HCP PMAT24 version?

It has been clarified that the two versions of the Penn Matrix Reasoning Test used to measure Pmat apply only to the PNC data in Results (p7; “because two versions of the Penn Matrix Reasoning Test were used to measure gF in the PNC...”), and a “PNC” label has been added to Supplementary Table 2. The differences in gF measures used in each data set have also been clarified in Methods (Cognitive prediction, p23):

In both data sets, fluid intelligence was quantified using matrix reasoning tests. In the HCP data set, a 24-item version of the Penn Progressive Matrices test was used; this test is an abbreviated form of Raven’s standard progressive matrices²⁰.

In the PNC data set, 24- and 18-item versions of the Penn Matrix Reasoning Test were used^{21,22}.

Fig. 2a, b. Can't read units on scales. Please enlarge font.

The colorbar font has been enlarged.

p. 11 para. 2. The FC measure is not directed, so please use 'patterns of connectivity with' rather than '.....to' here and elsewhere.

"To" has been replaced with "with" when discussing "patterns of connectivity," "connections," and regions that are "connected" with other regions.

Methods. Please state key parameters for HCP and PNC scans: Spatial resolution, multiband factor, TR, scan durations, to spare readers from needing to look these up.

Key imaging parameters have been added to Methods for both the HCP data set (Imaging parameters and preprocessing, p21: "In brief, all fMRI data were acquired on a 3T Siemens Skyra using a slice-accelerated, multiband, gradient-echo, echo planar imaging (EPI) sequence (TR = 720 ms, TE = 33.1 ms, flip angle = 52 degrees, resolution = 2.0 mm³, multiband factor = 8)") and the PNC data set (Imaging parameters and preprocessing, p22: "In brief, all fMRI data were acquired on a 3T Siemens TIM Trio using a multi-slice, gradient-echo EPI sequence (TR = 3,000 ms, TE = 32 ms, flip angle = 90 degrees, resolution = 3 mm³"). Scan durations for each condition have also been added to these sections.

Reviewer #2 (Remarks to the Author):

"Task-induced brain state manipulation improves prediction of individual traits" is an extension of the authors' previously published study on resting state fMRI fingerprints. Here they examine whether models built from task fMRI data perform better than those built from resting-state fMRI data. This a nice direction, but neither the results nor approach is novel, as several previous studies have examined this question (Cole et al and others).

While others have investigated the group-level similarities, differences, and transitions²³⁻³⁰ among "intrinsic" and task-related patterns of functional connectivity, to our knowledge, the few studies that have used task-based functional connectivity for prediction have focused on the prediction of state variables directly related to the given task (e.g.,^{31,32}). Our work demonstrates not only the existence of meaningful differences between rest- and task-based functional connectivity patterns, but also the utility of these differences to reveal brain-behavior relationships (see Introduction, p4 and Discussion, p15). That is, we show that task data can be used to improve prediction of stable traits, not only task-relevant states, suggesting an opportunity to use specific tasks to make meaningful behavioral and clinical predictions about individuals, and to deepen our understanding of the neural bases of such individual traits. These results thus provide strong motivation for a shift from resting-state functional connectivity to

task-based functional connectivity for the study of brain-behavior relationships, using thoughtfully selected tasks appropriate for the traits of interest.

Further, this study has several other major limitations: it is not theoretically motivated, tests no neural hypotheses, and the findings are incremental and inconsistent with respect to a previous publication from the authors (Finn et al., 2015 NN). Surprisingly, findings from that study are not discussed, although the authors employed almost exactly the same pipeline and investigated the same question “do individual connectivity profiles predict fluid intelligence”.

First, this study is intentionally data driven, a choice that permits identification of gF-relevant circuitry that would likely not be studied in a hypothesis-driven study (see response to Reviewer 2, comment 1 below). Second, the findings of this study are consistent with previous work from our lab^{4,5} and others^{33,34}: a significant amount of variance in gF is explained by gF-relevant network strength at rest. We note that the sample used by Finn et al.⁵ was smaller than the sample used in the current study, which likely contributed to better rest-based model performance in that study (see response to Reviewer 2, comment 2), but the main finding is replicated here. Finally, we emphasize that we applied connectome-based predictive modeling in this study not to further develop the pipeline, itself, which has already been extensively validated^{4,5,31,35}, but rather to demonstrate a key, generalizable point: cognitive tasks, by taxing trait-relevant neural circuitry, amplify individual differences in this circuitry and correspondingly permit more complete and robust characterization of brain-behavior relationships. This point has been clarified in the manuscript (see Discussion, p15).

Overall, the study is weak and findings generally uninterpretable. Other concerns include:

(1) There is no theoretical justification for using gambling, motor, social, and emotion tasks to predict fluid intelligence. Gambling task activation turned out to a strong predictor of gF. I wonder what the interpretation and theory here is.

The incentive processing, social cognition, and emotion processing tasks are complex tasks that activate broad swaths of the brain¹⁰, and it is unsurprising that motor task-based models perform relatively well given this task's robust activation of motor cortices¹⁰ and the finding that edges in the motor network tend to contribute disproportionately to gF predictive models (e.g., Fig. 2d; this is a discovery, we note, that likely would not have been made in a hypothesis-driven study). The consistently high performance of the incentive processing, or gambling, task-based model is perhaps also unsurprising given the high-level cognitive functions into which this task taps (e.g., reward processing, decision-making), the broad distribution of regions (both related and unrelated to reward) it activates^{10,36}, and the well-documented individual differences in striatal reward response that it elicits^{10,37}. Again, however, given the extensive literature highlighting the overlap between intelligence- and working memory-related networks (for a summary of this work, see³⁸), a hypothesis-driven study would likely have overlooked the incentive processing task, limiting our capacity to identify and understand the most effective perturbations for amplification of individual differences in gF-related circuitry. To clarify these points, a statement about the complexity of these tasks and the data-driven nature of this work has been added to Results (p5).

(2) As in Finn et al., the results demonstrate that connectivity profiles can predict fluid intelligence. However, if we compare Finn et al. Fig. 5a with current Fig. 1d, it appears that models built from the previous small sample actually outperform those from the larger sample used here. This suggests that predictive values of connectivity-features (or the effectiveness of the current pipeline) decreases with increasing sample size?

First, we highlight that, despite differences in sample size, feature-selection thresholds, and statistical procedures, this study replicates the core finding of several recent studies, including that of Finn and colleagues^{4,5,33}: patterns of functional connectivity at rest can be used to predict individual measures of gF. That this finding generalizes across large, independent samples with different age ranges (i.e., HCP and PNC) is further evidence of its robustness.

Nevertheless, we appreciate that the differences between results in this manuscript and that of Finn *et al.* may at first appear counterintuitive and confusing. These differences are likely explained, in large part, by the difference in sample size, as this reviewer suggests. The results of Finn *et al.* were based on a smaller sample ($n = 118$) than is used in this study ($n = 515$). Overfitting is well known to be more likely and problematic when the number of predictors is large and the number of samples, small³⁹, such that a smaller effect in a larger sample often reflects a more accurate estimate of the true effect size, rather than a failure to replicate a larger effect in a smaller sample. This conclusion – that a larger sample size has allowed us to more accurately estimate the percent of gF variance explained by functional connectivity strength in gF-related networks – is further supported by the consistency of these results with other recent work using patterns of functional connectivity to predict gF^{4,33}. We have revised the Discussion (p17-18) to clarify this issue.

Moreover, in Finn et al study, just using features from fronto-parietal networks achieved the same performance as using features from the whole brain (Fig. 5a vs. 5c). Although the authors did not test if this is the case in the current study, it appears that visuomotor regions are more highly predictive of fluid intelligence in the current study (Fig. 2d). Does this fit any cognitive theory or models of fluid intelligence? I doubt it.

We apologize for our conflation of the terms “predictive” and “overrepresented” in the manuscript, and note here that the goal of the localization analyses (Results, Model features are spatially distributed and overlapping, p11-13) was to identify regions that were overrepresented in gF-related circuitry; this does not require that these regions and networks be more predictive of gF than others, particularly given that we normalized results (i.e., cells in Fig. 2d) by network size, such that small networks may contribute few edges in absolute terms, but may be relatively overrepresented in a given model. We have replaced the term “predictive” throughout this section of the manuscript to clarify this point.

To demonstrate this distinction, we performed a virtual “lesion” analysis, removing from connectivity matrices all nodes in motor and visual networks; patterns of model performance were largely unchanged, as shown below (results presented as $100r^2$ for each model):

	Task	Feature-selection threshold		
		$P < 0.01$	$P < 0.005$	$P < 0.001$
HCP	Gam	11.8	10.7	10.5
	R1	0.6	2.9	3.0
	R2	0	1.1	0
	Lang	9.4	10.3	10.0
	Mot	8.0	8.8	9.4
	Rel	7.2	6.8	6.3
	Soc	7.0	6.1	5.1
	WM	11.0	11.1	10.3
	Emo	6.7	3.7	8.5
PNC	WM	10.1	10.6	12.2
	Emo	5.4	7.0	9.2
	Rest	4.7	4.8	3.6

Further, the finding of widely distributed gF-related networks is consistent with an extensive literature documenting the neural underpinnings of gF⁴⁰⁻⁴² and general intelligence^{38,43-47}. The overrepresentation of motor and visual regions in predictive models of gF is similarly unsurprising as, in most cases, tasks used in the HCP and PNC data sets are visually complex and require motor responses^{10,48}. Representations of task strategy, engagement, and performance may thus be expected to be found in visual and motor regions, and it is likely that individual differences in these processes relate to individual differences in gF. Discussion in the main text of how these results fit into previous efforts to characterize the neural underpinnings of intelligence has been expanded (Discussion, p18-19).

In sum, our findings are consistent with an extensive literature documenting the distributed networks underlying gF and reasonable given the nature of the tasks. This study's data-driven approach has allowed us to look beyond oversimplified neural accounts of gF to interrogate this distributed circuitry and access additional insights into meaningful individual differences in it.

(3) Use of a developmental PNC cohort with a large age range is problematic.

It is of course true that there are meaningful differences between the brains of adults and children, but these differences make our results all the more compelling. That is, by using such different samples, we take external validation to its logical extreme: that a model built in adults can be applied to children and adolescents, and vice versa, is a strong endorsement of the generalizability of these models and their relative performance (see Discussion, p18). Further, in both data sets, task-based models outperformed rest-based models, and emotion task-based models outperformed WM task-based models in females while WM task-based models outperformed emotion task-based models in males; this replication in independent, very different data sets indicates that these results are robust and generalizable. We are excited to present such sample-invariant results that can guide future efforts to reveal and study brain-behavior relationships in a wide range of populations.

(4) The authors employed a univariate approach to select features using connectivity features that show a significant correlation with fluid intelligence. This is highly circular and problematic, and a likely reason for the inflated findings in Finn et al. This is now quite problematic as the current results suggest a failure to replicate original Finn et al. findings.

The use of a single measure (here, gF) for feature selection and prediction is neither circular nor problematic in our analyses because we use cross-validation to ensure that data used for training and testing the models are kept strictly separate (see Discussion, p17). That is, features are selected and each model is built using gF scores and edge strengths in $n - 1$ subjects; this model is then tested in the unseen n^{th} subject. This standard, leave-one-out cross-validation approach to prediction is described in Methods (Cognitive prediction, p23-24). This separation is even more conservative – and the demonstration of generalizability more compelling⁴⁹ – in the cross-data set analyses (Methods, p25), which show that models, particularly those built using task data, generalize from one data set to another, entirely independent data set (Results, p10-11). These procedures explicitly avoid the “double-dipping”⁵⁰ that could inflate findings, as this reviewer suggests, and thus do not explain differences between these results and those presented by Finn *et al.*⁵ We note, again, that this work does replicate the main prediction results of Finn *et al.*, with several sample and analysis differences that likely account for decreased rest-based model performance in this work (see response to Reviewer 2, comment 2).

Reviewer #3 (Remarks to the Author):

Review of: Task-induced brain state manipulation improves prediction of individual traits

I read this article with great interest not least because I agree strongly with the central premise, which is that dynamic/task active connectivity should provide a better predictor of cognitive ability than resting state connectivity. This notion makes a great deal of sense, given that we see patterns of large-scale network activity/coherences during the performance of g-loaded tasks. The analyses and results have a number of important strength, however, there are also a number of key points that I believe if addressed would greatly strengthen the article.

1) The authors state that “By considering task-induced brain state and sex, the best-performing model explains over 20% of the variance in fluid intelligence scores, as compared to less than 5% of variance explained by rest-based models built using the whole sample.”

This is an intriguing result, particularly given the WM/emotional task dissociation. The replication of the task*gender interaction across the two studies is a strength. My main concern though is that the difference in variance explained when accounting for gender could relate to differences in cohort size. By definition, there must be fewer subjects in the trained datasets for the gender analyses, i.e., because the cohort has been split into two groups. Is it not the case that they will be more prone to overfit, and therefore, could give the illusion of explaining more variance? To be truly convincing, the trained model would

really need to be validated by application to a further dataset. I also was curious why only gender was examined as opposed to other factors (handedness, age, education level, etc?)

We chose to examine sex because it is a salient group feature that has received much attention in the human neuroimaging – and, more specifically, functional connectivity – literature. Sex was also a sound choice because it provides a natural measure for dividing the subject pool into only two subgroups, as opposed to other features, such as age or education level, that don't provide such clean separation, or that separate the sample into even smaller subgroups. We present these sex differences both because we believe them to be an important indication that the neural representations of fluid intelligence vary by sex, and – perhaps even more importantly – because they provide proof of the principle that predictive modeling may be most successful when appropriate models are defined for a given group. Finally, the sex difference work demonstrates that brain state manipulations may not all have the same efficacy for every group. A more complete characterization of the features by which groups should be defined to maximize model performance is certainly of interest, as is a method to identify such features in a data-driven manner, and we consider these to be important questions for future research (see Discussion, p20).

As discussed elsewhere in this response, we share this reviewer's concern about overfitting, and recognize that any decrease in sample size increases the risk of overfitting³⁹, but we offer several key observations that mitigate this concern in the sex differences analysis. First, even after splitting the samples by sex, the groups are still quite large (each includes over 200 subjects). Second, with greater overfitting, average model performance would be expected to improve; this is not the case when the sample is split by sex (HCP whole sample mean $r^2 = 7.1\%$ versus HCP split sample mean $r^2 = 6.4\%$; PNC whole sample mean $r^2 = 8.7\%$ versus PNC split sample mean $r^2 = 6.8\%$). This suggests that overfitting is not more problematic in the sex differences analysis than in the whole-sample analysis. Finally, this analysis demonstrates a meaningful condition-by-sex interaction, but we make no claims about absolute model performance (see response to Reviewer 1, comment 8). If differences in model performance when the sample is split by sex were attributable to overfitting, alone, all models would demonstrate comparable changes in performance, as there is no reason to expect that overfitting would affect one condition more than others, and the pattern of model performance across conditions would be preserved and similar in males and females. This is not the case (i.e., the relative performance of emotion and WM task-based models in males is opposite that in females), further evidence that the sex difference in which brain state most improves gF prediction is not the product of overfitting. We do, however, agree with this reviewer that it will be productive and interesting to explore this effect in additional data sets, and we look forward to doing so in the future.

2) The main finding is that “brain state can be manipulated to better reveal brain-behavior relationships, and that identifying and inducing the right brain state in a given group can improve trait prediction”.

This seems sensible to me, g-loaded tasks involve activity/coherence across certain networks, it makes sense to examine those networks when they are expressing those active/synchronised states. The fact that “Results generalize across conditions and two

large, independent datasets” is also a strength - the importance of reproducibility is finally coming to the fore in the imaging field at the moment, which is a good thing.

We thank the reviewer for these supportive comments.

A concern though, is how well balanced this comparison actually is? Specifically, were the rest and task acquisitions the same duration? I.e., did they have the same acquisition parameters, and were there the same type and number of EPI scans for task and rest? This is an important point to consider. Longer acquisition will allow for a more robust estimate of correlation strength between each pair of nodes. this results in a connectivity matrix that has greater signal vs noise. One should ensure that the rest vs task difference is not simply a consequence of differences in the reliability of these estimates.

Methods (Imaging parameters and preprocessing, p21-22) have been revised to clarify that all fMRI data in a given data set were acquired using the same imaging protocol, and relevant imaging parameters^{48,51} have been added to Methods (Imaging parameters and preprocessing, p21-22). Condition duration did differ for each condition; lengths of each condition have been added to Methods (p21-22). In the HCP data, rest runs were substantially longer than task runs (WM, 5:01; gambling, 3:12; motor, 3:34; language, 3:57; social, 3:27; relational, 2:56; emotion, 2:16; rest, 14:33), while in the PNC data, the opposite was true (WM, 11:39; emotion, 10:36; rest, 6:18). We appreciate that condition duration may affect the reliability of functional connectivity measures and, in turn, the success of predictive modeling, and note that, for this reason, the poor performance of HCP rest-based models is particularly impressive given the longer duration of the rest runs (Discussion, p15). To further investigate the potential effects of condition duration on model performance, we performed several additional analyses, described in detail in our response to Reviewer 1, comment 3; we found no clear relationship between condition duration and resulting model performance, and task-based models still outperformed rest-based models after connectivity matrices were recalculated using the same number of frames from each condition. In the manuscript, these analyses are described in Methods (Effects of condition duration, p26-27), and corresponding results are presented in Supplementary Table 6 and summarized in Results (Investigation of potential confounds, p8). We also investigated a potential relationship between reliability at the node level and model node degree, and found no significant relationship (see response to Reviewer 1, comment 7).

3) Relatedly to the above two points, when stating how much variance is actually explained, I would have been more convinced if the headline value was for a cross validation. by this, I mean where the model is trained on HCP data for one task, and then validated with more HCP data for that same task. I don't question that there is a connectivity-g relationship, but without such a step, the headline estimates of actual variance explained are hard to interpret (I note, that this is a criticism that can be levelled at a number of other high impact papers, and the authors at least apply a cross validation across studies, albeit with somewhat different tasks/acquisitions).

We agree that cross-validation is critical to avoid overfitting and stringently test model generalizability, and clarify that, for this reason, we used a leave-one-out cross-validation approach when training and testing all CPM models, with the exception of the cross-data set

analysis, such that data used for training and testing were always kept strictly separate. This ensures that all prediction results presented in the manuscript are cross-validated. These methods are described in full in Methods (Cognitive prediction, p23-24 and Validation of the models, p24-25). We also recognize the benefits and potential pitfalls of leave-one-out and k -fold cross-validation approaches, and have re-run our analyses using k -fold cross-validation for model training and testing. This analysis and corresponding results are discussed below (see response to Reviewer 3, comment 8).

4) The comparison of rest vs. task seems well balanced and sensible (assuming the above condition regarding number of TRs is met). A question is whether the authors also looked at measures taken of dynamic connectivity relative to steady state (PPI for example), or were the average taken across both rest and task for the task acquisitions? In a way it is critical to use the simple correlational as opposed to PPI approach, as this is more balanced across task and rest conditions, making them more comparable. Nonetheless, an obvious prediction of the authors hypothesis is that the PPI measures (increased connectivity during task vs rest) should in fact give the best possible explanation of g.

We apologize for the confusion regarding our approach and note that we chose to use the simple correlational approach – that is, correlating time courses across entire task runs without accounting for task design – for several reasons: to make models derived from task and rest conditions more comparable, as suggested by this reviewer; to permit identification of stable, core gF-related circuitry in all conditions (including at rest; PPI would ignore any such connections that fail to meaningfully change across conditions, i.e., connections that are part of an “intrinsic” functional architecture²³); and to treat each task as a separate and continuous brain state.

We are also interested in the performance of PPI measures, with the caveat, as noted above, that such measures may fail to capture stable gF-related circuitry; nevertheless, this remains an important question for future work, and we look forward to investigating it further to more comprehensively and mechanistically characterize task-induced brain states.

5) I was somewhat confused by the description of supplementary analyses with more robust compensation for movement artefacts. Does this mean that movement still had a component in the connectivity-g relationship for the primary analyses as reported in the main text? if so, then really these supplementary analyses should be presented as the main analyses, and headline statistics altered accordingly. If not, and the minimal data cleaning pipeline was sufficient, then are these not redundant?

While we employed a conservative threshold for motion exclusion in both data sets (mean frame-to-frame displacement less than 0.1 mm, and maximum frame-to-frame displacement less than 0.15 mm), gF remained correlated with motion in 3 out of 21 runs (Methods, p21-22). Given this, as well as the finding that model predictions were in many cases correlated with mean frame-to-frame displacement, we were concerned that motion could be confounding the primary analyses, as this reviewer suggests. To ensure that this was not the case, we undertook several additional analyses (incorporation of mean frame-to-frame displacement into feature-selection and model building steps; Methods, Effects of head motion, p25-26). We predicted that these

analyses would be redundant, indicating that the variance in true and predicted gF that can be explained by motion is relatively non-overlapping with the variance in these measures explained by network strength. This was indeed the case, as model performance was not substantially changed by these additional analyses (Supplementary Table 4). In sum, we agree that these analyses are redundant, but feel that this redundancy is important to highlight, as it addresses concerns that models may be based on or predictive of participant motion, rather than gF; that is, these supplementary analyses indicate that motion is not a meaningful confound in our primary analyses.

6) The observation that models trained on rest data provide stronger prediction of ‘g’ when applied to task data seems to be a strength - although I was not clear on how generally / consistently this was the case - it would be good to clarify this in the text. Also, to clarify that the task and rest acquisitions actually have the same amount of data as outlined above.

We have clarified in the text that in ten out of twelve tested cases, models trained on rest data and tested on WM task data outperformed models trained and tested on rest data:

In fact, in all but 2 out of 12 tested cases, the rest-based models performed better when applied to the WM data than when applied to the rest data on which they were built (Results, p10).

See responses to Reviewer 3, comment 2 and Reviewer 1, comment 3 for a discussion of the relationship between condition duration and model performance.

7) Repeating analyses with different parcellation resolutions is a strength of the paper.

We thank the reviewer for this supportive comment.

8) It is surprising that leave-one-out approach was applied in such a large dataset - the gold standard would be to use cross validation with train and test sub-groups.

We appreciate the bias-variance trade-off inherent in selecting a cross-validation approach³⁹; to empirically address this concern, we re-ran main analyses using k -fold cross-validation, with $k = 10$, for both the HCP and PNC data sets. This analysis approach is described in Methods (Effects of cross-validation method, p27); corresponding results ($100r^2$) are presented in Supplementary Table 7, reproduced below, and summarized in Results (Investigation of potential confounds, p8). Of note, these results demonstrate no substantial differences in absolute or relative model performance compared to models generated using a leave-one-out cross-validation approach.

	Task	Feature-selection threshold		
		$P < 0.01$	$P < 0.005$	$P < 0.001$
HCP	Gam	13.0	12.9	13.0
	R1	3.4	4.2	4.9
	R2	1.0	0.4	0
	Lang	8.1	8.2	8.4
	Mot	8.3	8.2	8.3
	Rel	4.8	4.8	4.8
	Soc	8.7	8.6	6.8
	WM	11.4	11.3	10.0
	Emo	6.7	6.9	7.0
PNC	WM	10.4	11.0	10.8
	Emo	8.8	9.3	9.7
	Rest	4.4	4.3	3.9

9) I was unclear whether the number of edges that were feature selected in the male and female populations was balanced? Obviously any differences in this would lead to a problem, whereby more edges gives the model more degrees of freedom for fitting (and over-fitting) the data.

We thank the reviewer for this important question, and note that it touches upon several points of great interest to us.

First, we demonstrate in the main analyses that P value-based thresholds and sparsity thresholds for feature-selection yield models with comparable absolute and relative performance (Supplementary Table 1). Additionally, when considering models generated using P value-based feature-selection thresholds, we find that better-performing models include more edges (Supplementary Fig. 2), a finding that lends further support to the idea that task-induced brain states improve gF prediction by amplifying and revealing individual differences in patterns of functional connectivity, such that more edges are significantly related to gF and thus leveraged for its prediction. We therefore suggest that allowing models to include varying numbers of edges is informative, and present analyses using P value-based thresholds in the manuscript and figures (for both whole-sample and sex differences analyses).

As discussed above, we are generally concerned and careful about potential overfitting, but because models are trained and tested on summary statistics – that is, on unweighted sums of selected edges’ strengths – the number of edges included in the model should not change the model’s degrees of freedom or the likelihood of overfitting.

Nevertheless, we believe that it is important to employ sparsity thresholds as a control to ensure that the number of edges included in the models does not have any unforeseen effects on model performance. It is for this reason that we used both P value-based and sparsity thresholds in the main analyses, and in the same spirit, we re-ran CPM on males and females separately using sparsity, rather than P value-based, thresholds. Below, we present results (i.e., $100r^2$) for the sparsity thresholds that select numbers of edges closest to those selected using a feature-selection

threshold of $P < 0.01$. For comparison, we also present results of models generated using a feature-selection threshold of $P < 0.01$ (as presented in the manuscript), and note that the same trends hold (i.e., WM task-based models outperformed emotion task-based models in males, while the opposite was true in females):

	Task	$P < 0.01$		Top 1%		Top 2.5%	
		M	F	M	F	M	F
HCP	Gam	14.4	7.6	15.0	6.8	13.8	7.2
	R1	0.7	0.1	5.0	0	1.9	0
	R2	1.9	0.4	2.7	0	2.2	0.9
	Lang	6.2	7.9	6.2	7.8	5.5	8.4
	Mot	9.0	5.3	9.0	4.7	11.5	5.0
	Rel	3.8	5.9	3.6	5.7	4.1	4.0
	Soc	14.6	4.2	13.3	3.6	14.7	3.0
	WM	20.3	0.5	15.5	0.5	21.3	3.4
	Emo	7.3	5.9	8.5	6.2	9.0	5.0
PNC	WM	9.7	6.3	10.9	4.6	9.3	8.0
	Emo	4.0	11.8	4.6	12.0	3.4	12.3
	Rest	5.3	3.7	5.4	5.0	6.2	6.7

In summary this is an intriguing and potentially important paper with great potential. I hope that the above suggestions are helpful in strengthening it further.

Additional Note

We note one additional change to the manuscript: in the process of revising its contents, we discovered that there were several HCP subjects lacking coverage in 9/268 nodes. As in the PNC data set, we adopted the conservative approach of excluding these nodes from all subjects. We re-ran all analyses with these nodes excluded and revised the manuscript accordingly; there were no substantial changes in revised results or their interpretation.

Once again, we thank the reviewers for their helpful comments, and look forward to continuing to explore many of these important ideas.

References

1. Winkler, A. M., Webster, M. A., Vidaurre, D., Nichols, T. E. & Smith, S. M. Multi-level block permutation. *NeuroImage* **123**, 253–268 (2015).
2. Winkler, A. M., Ridgway, G. R., Webster, M. A., Smith, S. M. & Nichols, T. E. Permutation inference for the general linear model. *NeuroImage* **92**, 381–397 (2014).
3. Benjamini, Y. & Hochberg, Y. Controlling the false discovery rate: a practical and powerful approach to multiple testing. *J. R. Stat. Soc. B* **57**, 289–300 (1995).
4. Noble, S. *et al.* Influences on the test–retest reliability of functional connectivity MRI and its relationship with behavioral utility. *Cereb. Cortex* **27**, 5415–5429 (2017).
5. Finn, E. S. *et al.* Functional connectome fingerprinting: identifying individuals using patterns of brain connectivity. *Nat. Neurosci.* **18**, 1664–1671 (2015).
6. Power, J. D. *et al.* Methods to detect, characterize, and remove motion artifact in resting

- state fMRI. *NeuroImage* **84**, 320–341 (2014).
7. Marrelec, G. *et al.* Partial correlation for functional brain interactivity investigation in functional MRI. *NeuroImage* **32**, 228–237 (2006).
 8. Smith, S. M. *et al.* Network modelling methods for FMRI. *NeuroImage* **54**, 875–891 (2011).
 9. Wang, Y., Kang, J., Kemmer, P. B. & Guo, Y. An efficient and reliable statistical method for estimating functional connectivity in large scale brain networks using partial correlation. *Front. Neurosci.* **10**, (2016).
 10. Barch, D. M. *et al.* Function in the human connectome: task-fMRI and individual differences in behavior. *NeuroImage* **80**, 169–189 (2013).
 11. Buckner, R. L., Krienen, F. M. & Yeo, B. T. T. Opportunities and limitations of intrinsic functional connectivity MRI. *Nat. Neurosci.* **16**, 832–837 (2013).
 12. Elton, A. & Gao, W. Task-related modulation of functional connectivity variability and its behavioral correlations. *Hum. Brain Mapp.* **36**, 3260–3272 (2015).
 13. Hasson, U., Nusbaum, H. C. & Small, S. L. Task-dependent organization of brain regions active during rest. *Proc. Natl. Acad. Sci. U. S. A.* **106**, 10841–10846 (2009).
 14. Tailby, C., Masterton, R. A. J., Huang, J. Y., Jackson, G. D. & Abbott, D. F. Resting state functional connectivity changes induced by prior brain state are not network specific. *NeuroImage* **106**, 428–440 (2015).
 15. Gregory, M. D., Robertson, E. M., Manoach, D. S. & Stickgold, R. Thinking about a task is associated with increased connectivity in regions activated by task performance. *Brain Connect.* **6**, 164–168 (2016).
 16. Lowe, M. J., Dzemidzic, M., Lurito, J. T., Mathews, V. P. & Phillips, M. D. Correlations in low-frequency BOLD fluctuations reflect cortico-cortical connections. *NeuroImage* **12**, 582–587 (2000).
 17. Leonardi, N., Shirer, W. R., Greicius, M. D. & Van De Ville, D. Disentangling dynamic networks: separated and joint expressions of functional connectivity patterns in time. *Hum. Brain Mapp.* **35**, 5984–5995 (2014).
 18. Salehi, M., Karbasi, A., Scheinost, D. & Constable, R. T. A submodular approach to create individualized parcellations of the human brain. In: Descoteaux, M. *et al.* (eds) *Medical Image Computing and Computer Assisted Intervention – MICCAI 2017. Lecture Notes in Computer Science, vol 10433* 478–485 (Springer, Cham, 2017).
 19. Glasser, M. F. *et al.* A multi-modal parcellation of human cerebral cortex. *Nature* **536**, 171–178 (2016).
 20. Bilker, W. B. *et al.* Development of abbreviated nine-item forms of the Raven’s standard progressive matrices test. *Assessment* **19**, 354–369 (2012).
 21. Moore, T. M., Reise, S. P., Gur, R. E., Hakonarson, H. & Gur, R. C. Psychometric properties of the Penn Computerized Neurocognitive Battery. *Neuropsychology* **29**, 235–246 (2015).
 22. Gur, R. C. *et al.* A cognitive neuroscience-based computerized battery for efficient measurement of individual differences: standardization and initial construct validation. *J. Neurosci. Methods* **187**, 254–262 (2010).
 23. Cole, M. W., Bassett, D. S., Power, J. D., Braver, T. S. & Petersen, S. E. Intrinsic and task-evoked network architectures of the human brain. *Neuron* **83**, 238–251 (2014).
 24. Lowe, M. J., Mock, B. J. & Sorenson, J. A. Functional connectivity in single and multislice echoplanar imaging using resting-state fluctuations. *NeuroImage* **7**, 119–132

- (1998).
25. Krienen, F. M., Yeo, B. T. T. & Buckner, R. L. Reconfigurable task-dependent functional coupling modes cluster around a core functional architecture. *Philos. Trans. R. Soc. Lond. B. Biol. Sci.* **369**, (2014).
 26. Cole, M. W. *et al.* Multi-task connectivity reveals flexible hubs for adaptive task control. *Nat. Neurosci.* **16**, 1348–1355 (2013).
 27. Rissman, J., Gazzaley, A. & D’Esposito, M. Measuring functional connectivity during distinct stages of a cognitive task. *NeuroImage* **23**, 752–763 (2004).
 28. Shirer, W. R., Ryali, S., Rykhlevskaia, E., Menon, V. & Greicius, M. D. Decoding subject-driven cognitive states with whole-brain connectivity patterns. *Cereb. Cortex* **22**, 158–165 (2012).
 29. Greicius, M. D., Krasnow, B., Reiss, A. L. & Menon, V. Functional connectivity in the resting brain: a network analysis of the default mode hypothesis. *Proc. Natl. Acad. Sci.* **100**, 253–258 (2003).
 30. Mennes, M., Kelly, C., Colcombe, S., Castellanos, F. X. & Milham, M. P. The extrinsic and intrinsic functional architectures of the human brain are not equivalent. *Cereb. Cortex* **23**, 223–229 (2013).
 31. Rosenberg, M. D. *et al.* A neuromarker of sustained attention from whole-brain functional connectivity. *Nat. Neurosci.* **19**, 165–171 (2016).
 32. Ashar, Y. K., Andrews-Hanna, J. R., Dimidjian, S. & Wager, T. D. Empathic care and distress: predictive brain markers and dissociable brain systems. *Neuron* **94**, 1263–1273 (2017).
 33. Dubois, J., Galdi, P., Han, Y., Paul, L. K. & Adolphs, R. Predicting personality traits from resting-state fMRI. Preprint at <https://www.biorxiv.org/content/early/2017/11/07/215129> (2017).
 34. Santarnecchi, E. *et al.* Network connectivity correlates of variability in fluid intelligence performance. *Intelligence* **65**, 35–47 (2017).
 35. Shen, X. *et al.* Using connectome-based predictive modeling to predict individual behavior from brain connectivity. *Nat. Protoc.* **12**, 506–518 (2017).
 36. Delgado, M. R., Nystrom, L. E., Fissell, C., Noll, D. C. & Fiez, J. A. Tracking the hemodynamic responses to reward and punishment in the striatum. *J. Neurophysiol.* **84**, 3072–3077 (2000).
 37. Hariri, A. R. *et al.* Preference for immediate over delayed rewards is associated with magnitude of ventral striatal activity. *J. Neurosci.* **26**, 13213–13217 (2006).
 38. Colom, R., Karama, S., Jung, R. E. & Haier, R. J. Human intelligence and brain networks. *Dialogues Clin. Neurosci.* **12**, 489–501 (2010).
 39. Yarkoni, T. & Westfall, J. Choosing prediction over explanation in psychology: lessons from machine learning. *Perspect. Psychol. Sci.* **12**, 1100–1122 (2017).
 40. Prabhakaran, V., Smith, J. A. L., Desmond, J. E., Glover, G. H. & Gabrieli, J. D. E. Neural substrates of fluid reasoning: an fMRI study of neocortical activation during performance of the Raven’s Progressive Matrices Test. *Cogn. Psychol.* **33**, 43–63 (1997).
 41. Choi, Y. Y. *et al.* Multiple bases of human intelligence revealed by cortical thickness and neural activation. *J. Neurosci.* **28**, 10323–10329 (2008).
 42. Ebisch, S. J. *et al.* Common and unique neuro-functional basis of induction, visualization, and spatial relationships as cognitive components of fluid intelligence. *NeuroImage* **62**, 331–342 (2012).

43. Duncan, J. *et al.* A neural basis for general intelligence. *Science* **289**, 457–460 (2000).
44. Barbey, A. K. *et al.* An integrative architecture for general intelligence and executive function revealed by lesion mapping. *Brain* **135**, 1154–1164 (2012).
45. Jung, R. E. & Haier, R. J. The Parieto-Frontal Integration Theory (P-FIT) of intelligence: converging neuroimaging evidence. *Behav. Brain Sci.* **30**, 135–154 (2007).
46. Colom, R., Jung, R. E. & Haier, R. J. Distributed brain sites for the g-factor of intelligence. *NeuroImage* **31**, 1359–1365 (2006).
47. Colom, R. *et al.* Gray matter correlates of fluid, crystallized, and spatial intelligence: testing the P-FIT model. *Intelligence* **37**, 124–135 (2009).
48. Satterthwaite, T. D. *et al.* Neuroimaging of the Philadelphia Neurodevelopmental Cohort. *NeuroImage* **86**, 544–553 (2014).
49. Rosenberg, M. D., Finn, E. S., Scheinost, D., Constable, R. T. & Chun, M. M. Characterizing attention with predictive network models. *Trends Cogn. Sci.* **21**, 290–302 (2017).
50. Kriegeskorte, N., Simmons, W. K., Bellgowan, P. S. F. & Baker, C. I. Circular analysis in systems neuroscience: the dangers of double dipping. *Nat. Neurosci.* **12**, 535–40 (2009).
51. Uğurbil, K. *et al.* Pushing spatial and temporal resolution for functional and diffusion MRI in the Human Connectome Project. *NeuroImage* **80**, 80–104 (2013).

Reviewers' comments:

Reviewer #1 (Remarks to the Author):

The authors have been very responsive to the issues raised by all three reviewers, and the manuscript is substantially improved. I have one comment that may warrant additional analysis:

In Fig. 2c, visual inspection of the complex spatial patterns suggests a reasonable degree of bilateral symmetry. It would be very informative if this could be quantified. This might be challenging if there are not good node correspondences between the left and right hemispheres in the 268-node parcellation. But given the high degree of bilateral symmetry in functional organization, it is important to know how symmetric are these results.

The other comments are relatively minor:

- 1) The black square in Fig. 1i was confusing until I realized it signified four solid colors. Can it be converted to a square made up of four appropriately colored triangles?
- 2) P. 12, fig. 2d. This figure shows the fraction of edges belonging to different internetwork pairs. But since there are 268 nodes and 10 networks, that's 27 nodes/network. What fraction of the edges in this analysis are intra-network and not inter-network? If that's been analyzed/reported, I missed it (but perhaps didn't read the methods closely enough).
- 3) Line 271. Can the 10 networks be shown in a supplemental figure for easy reference? If not, please point to the specific figure in an earlier publication.
- 4) Para. starting line 276. Include the network numbers in the main text when describing 'visual' etc. so the reader isn't forced to jump to the legend to follow the description of fig. 2d.
- 5) Line 394 "some overfitting is almost inevitable" Is that really true? Would it be better to say "overfitting is difficult to eliminate completely"?
- 6) Line 545 and Figs S3 and S4. The statement that 'the scan volume was too restricted' is puzzling. Fig. S4 seems to imply that the HCP fMRI data lacked full coverage of the cerebellum, but that is not the case. This should be clarified/corrected so that readers don't get an incorrect impression and so that the actual explanation is more clear.

Reviewer #2 (Remarks to the Author):

The authors have done a good job of addressing the several detailed issues raised in the previous revision. However, several major concerns remain as noted below.

1. In terms of overall claims and rationale, I appreciate the sentiment that "...there is a strong motivation for a shift from resting-state functional connectivity to task-based functional connectivity for the study of brain-behavior relationships". But relating task-based functional connectivity to cognition and behavior, and predicting them, are precisely what researchers have been doing for over 15 years before resting-state fMRI turned it into a cottage industry correlating everything with everything in the most atheoretical ways imaginable! Are we now not just circling back albeit with more brain features – some justified and many largely not? As such, the use of "paradigm shift" in the Abstract and manuscript is not warranted.
2. It appears that key aspects, especially feature selection in the previous study (Finn et al. 2015), could not be replicated here. This is a major concern in our field, and especially so for claims of predictive modeling. The authors attributed better rest-based model in Finn et al to overfitting but I remain concerned that the same issues may hold for the present study because the methods used may not be robust or stable. It would be important to do a proper validation study with multiple repeats and stability analysis. E.g. randomly split the data into subsets (e.g., 258 subjects in training

set and 257 subjects in testing set) and do prediction, feature selection analyses and repeat this procedure several (~ 1000) times to test for robustness and stability.

3. The use of the CV procedures is problematic for univariate feature selection. It is important not to claim these as predictive features and the manuscript needs to be revised accordingly.

4. Interpretation of features remains problematic: the authors' justification for using gambling, motor, social and emotion tasks to predict fluid intelligence is still weak, and this aspect has not been improved. The authors have not discussed what the features mean and how they contribute to fluid intelligence. Instead, they note that it is unsurprising that e.g. motor task-based models perform relatively well. Similarly, explaining fluid intelligence from gambling task features is not quite meaningful or interpretable. If the goal is solely prediction and the features are essentially uninterpretable, why discuss the features and tasks at such length?

Reviewer #3 (Remarks to the Author):

The authors have addressed all of my comments thoroughly. I have no further suggestions.

We thank the reviewers for these additional comments, which have motivated analyses and revisions that we believe substantially strengthened the manuscript. These revisions are described below, and are highlighted in the manuscript using blue text.

Reviewer #1 (Remarks to the Author):

The authors have been very responsive to the issues raised by all three reviewers, and the manuscript is substantially improved. I have one comment that may warrant additional analysis:

In Fig. 2c, visual inspection of the complex spatial patterns suggests a reasonable degree of bilateral symmetry. It would be very informative if this could be quantified. This might be challenging if there are not good node correspondences between the left and right hemispheres in the 268-node parcellation. But given the high degree of bilateral symmetry in functional organization, it is important to know how symmetric are these results.

We thank the reviewer for this interesting question. To quantify model symmetry while accounting for parcellation asymmetry, we used the centroid of each node to assign it to its homolog in the opposite hemisphere. That is, for each node, we calculated the distance between its centroid and the centroids of all nodes in the opposite hemisphere after reflecting them over the midline; the node in the reflected hemisphere that was closest to the given node was assigned as its match. This procedure was performed for all nodes in the right hemisphere (i.e., matching them with nodes in the reflected left hemisphere; “Right to Left”), and again for all nodes in the left hemisphere (i.e., matching them with nodes in the reflected right hemisphere; “Left to Right”). Nodes that lacked coverage – along with their paired nodes – were excluded from further analyses. Degree values of nodes in a given hemisphere were correlated with degree values of their assigned homologs using Spearman’s correlation for both the correlated and anti-correlated networks (CN and AN, respectively). Correlations indicate substantial bilateral symmetry of degree maps, as predicted. This symmetry is increased in models that yielded more accurate predictions of gF, further suggesting better resolution of gF-relevant circuitry using task- than rest-based functional connectivity. A description of this analysis has been added to Methods (p28-29), and results are summarized on p11-12 and presented in Supplementary Table 9 (reproduced below):

		CN		AN	
Task		Right to Left	Left to Right	Right to Left	Left to Right
PNC	WM	0.55 (5.0E-11)	0.56 (7.4E-12)	0.57 (8.6E-12)	0.62 (1.4E-14)
	Emotion	0.57 (6.3E-12)	0.54 (1.1E-10)	0.51 (2.5E-9)	0.45 (1.8E-7)
	Rest	0.29 (0.001)	0.31 (3.7E-4)	0.38 (1.3E-5)	0.40 (4.0E-6)
HCP	WM	0.32 (2.3E-4)	0.40 (3.2E-6)	0.38 (1.3E-5)	0.37 (1.9E-5)
	Emotion	0.27 (0.002)	0.27 (0.002)	0.32 (2.1E-4)	0.31 (3.3E-4)
	Rest1	0.17 (0.052)	0.19 (0.034)	0.31 (3.4E-4)	0.30 (6.5E-4)

Results presented as $r(P)$; models generated using an edge-selection threshold of $P < 0.01$ for consistency with main degree analyses.

The other comments are relatively minor:

1) The black square in Fig. 1i was confusing until I realized it signified four solid colors. Can it be converted to a square made up of four appropriately colored triangles?

The figure legend has been modified accordingly.

2) P. 12, fig. 2d. This figure shows the fraction of edges belonging to different internetwork pairs. But since there are 268 nodes and 10 networks, that's 27 nodes/network. What fraction of the edges in this analysis are intra-network and not inter-network? If that's been analyzed/reported, I missed it (but perhaps didn't read the methods closely enough).

We thank the reviewer for raising the important point of differences in network size; while we reference these differences in the text, we had not previously reported the total number of nodes in each canonical network. For concision and clarity, the number of nodes per network, as well as the number of nodes remaining in each network after node exclusion for each data set, are now presented in Supplementary Table 11 (reproduced below). The number of intra-network and inter-network edges for each network pair can be inferred from this table. Fig. 2d shows the relative contribution of each network pair (accounting for network sizes); contributions of intra-network edges are represented along each main diagonal, while contributions of inter-network edges are represented in off-diagonal elements.

Canonical network	Number of nodes		
	Overall	HCP	PNC
MF	29	29	29
FP	34	34	31
DMN	20	18	18
Motor	50	49	46
Vis A	18	18	18
Vis B	9	9	8
Vis Assoc	18	18	17
Saliency	30	30	30
Subcortical	29	29	29
CBL	31	25	24

3) Line 271. Can the 10 networks be shown in a supplemental figure for easy reference? If not, please point to the specific figure in an earlier publication.

A visualization of the 10 networks has been added as Supplementary Figure 6.

4) Para. starting line 276. Include the network numbers in the main text when describing 'visual' etc. so the reader isn't forced to jump to the legend to follow the description of fig. 2d.

The corresponding network numbers have been added after each mention of a canonical network (p13).

5) Line 394 “some overfitting is almost inevitable⁵¹” Is that really true? Would it be better to say “overfitting is difficult to eliminate completely”?

We thank the reviewer for pointing out this overstatement and suggesting an alternative way to express this idea. The phrase has been revised as suggested (p18).

6) Line 545 and Figs S3 and S4. The statement that ‘the scan volume was too restricted’ is puzzling. Fig. S4 seems to imply that the HCP fMRI data lacked full coverage of the cerebellum, but that is not the case. This should be clarified/corrected so that readers don’t get an incorrect impression and so that the actual explanation is more clear.

In a relatively small subset of subjects in each data set, the scan volume was limited such that some of the nodes in the parcellation lacked coverage. We adopted the conservative approach of excluding a node from all subsequent analyses of a given data set if it lacked coverage in *any* subject in that data set. We have revised the manuscript to clarify this point: “Of note, in a subset of subjects in each data set, some of these nodes lacked sufficient coverage (the scan volume was too restricted); we adopted the conservative approach of excluding these nodes in all subjects. In the HCP data, 9 nodes lacked sufficient coverage, and were dropped from all further HCP analyses. Nine additional nodes lacked sufficient coverage in the PNC data (for a total of 18 nodes with incomplete coverage in the PNC data); these 18 nodes were dropped from all further PNC and cross-data set analyses. These nodes were primarily in subcortical regions (Supplementary Figs. 2 and 3).” (p23).

Reviewer #2 (Remarks to the Author):

The authors have done a good job of addressing the several detailed issues raised in the previous revision. However, several major concerns remain as noted below.

1. In terms of overall claims and rationale, I appreciate the sentiment that “...there is a strong motivation for a shift from resting-state functional connectivity to task-based functional connectivity for the study of brain-behavior relationships”. But relating task-based functional connectivity to cognition and behavior, and predicting them, are precisely what researchers have been doing for over 15 years before resting-state fMRI turned it into a cottage industry correlating everything with everything in the most atheoretical ways imaginable! Are we now not just circling back albeit with more brain features – some justified and many largely not? As such, the use of “paradigm shift” in the Abstract and manuscript is not warranted.

While we are not the first to relate task-based functional connectivity to behavior, we note several key features that distinguish this work. First, we use patterns of functional connectivity to predict – rather than explain – individual traits; this rigorous approach to identifying brain-behavior relationships ensures that models are more robust and generalizable than explanatory models¹. Second, we use whole-brain, data-driven analysis techniques in recognition of the complexity and distributed nature of neural representations of high-level cognitive processes², such as those reflected in fluid intelligence scores (which has not been done before). Third, and

most importantly, we subvert the current trend to perform such analyses using rest-based functional connectivity^{2,3}, demonstrating that task-induced functional connectivity changes amplify individual differences in brain organization that are related to individual differences in fluid intelligence. This finding is replicated in two independent data sets, holds for two additional measures (PVRT and WRAT), and is robust to data processing choices, suggesting its generalizability. In short, we present the novel idea that whole-brain, task-based functional connectivity is better suited to the study of individual differences in brain organization (and related individual differences in cognition and behavior) than rest-based functional connectivity. Given the widespread use of resting-state functional connectivity to study brain-behavior relationships, we maintain our conviction that this finding suggests a paradigm shift in functional connectivity analyses. We highlight in the abstract that we are specifically exploring the utility of task-based functional connectivity for the study of brain-behavior relationships, and have clarified the last sentence of the abstract to reflect the broad relevance of these findings to functional connectivity analyses.

2. It appears that key aspects, especially feature selection in the previous study (Finn et al. 2015), could not be replicated here. This is a major concern in our field, and especially so for claims of predictive modeling. The authors attributed better rest-based model in Finn et al to overfitting but I remain concerned that the same issues may hold for the present study because the methods used may not be robust or stable. It would be important to do a proper validation study with multiple repeats and stability analysis. E.g. randomly split the data into subsets (e.g., 258 subjects in training set and 257 subjects in testing set) and do prediction, feature selection analyses and repeat this procedure several (~ 1000) times to test for robustness and stability.

We share this reviewer's concerns about overfitting, and have already taken several steps to ensure that patterns of model performance are reliable, that models are generalizable, and that model anatomy is relatively stable. First, k -fold cross-validation approaches yield results that are comparable to those generated by leave-one-out cross-validation approaches (see Supplementary Tables 1 and 7). Second, models built from one data set can be applied to another; given the differences between the HCP and PNC samples and experimental designs, this external validation is particularly compelling evidence of model robustness and generalizability. Third, models overlap across conditions and data sets, demonstrating stability of model anatomy (e.g., Fig. 2).

To further interrogate model overlap and ensure that it extends to confound analyses, we calculated the Spearman's correlations between model node degree in the main analyses, and model node degree in head motion analyses (i.e., edge selection via partial correlation with gF, controlling for motion; Supplementary Table 4) using an edge-selection threshold of $P < 0.01$. Correlations were quite high [see below; results reported as $r(P)$], indicating that hubs are robust to edge-selection approach and unrelated to head motion.

		WM	Emotion	Rest
HCP	CN	0.9680 (2.25E-156)	0.8899 (1.42E-89)	0.9841 (9.32E-195)
	AN	0.9747 (2.39E-169)	0.8728 (5.04E-82)	0.9845 (3.25E-196)
PNC	CN	0.9956 (2.64E-257)	0.9668 (5.51E-149)	0.9684 (1.28E-151)
	AN	0.9962 (2.56E-264)	0.9491 (1.79E-126)	0.9719 (7.73E-158)

Next, to evaluate the stability of these results, we performed the requested analysis; that is, we randomly divided the data set in half, trained the models on one half and tested on the other, and repeated this procedure 1,000 times. We performed this split-half analysis in both the HCP and PNC data sets and evaluated the stability of model performance (i.e., the correlation between predicted and true gF) and of node degree, a proxy for the stability of model anatomical distribution. To quantify the similarity of node degree across the 1,000 iterations, we calculated the Spearman's correlation between degree vectors for every pair of iterations. These additional analyses confirm that models' prediction performance and anatomical distribution are quite stable; analysis details have been added to Methods (p24 and p29) and results are presented in Supplementary Fig. 1 (reproduced below).

Supplementary Figure 1. Prediction performance (a,b) and model anatomical distribution (c,d) are relatively stable across 1,000 iterations of split-half predictive modeling in both the HCP (a,c) and PNC (b,d) data sets. Models generated using a feature-selection threshold of $P < 0.01$. Results presented as the Spearman's correlation between predicted and true gF (a,b) and between node degree vectors for every pair of iterations. In each boxplot, center line corresponds to the median value, box edges correspond to the 25th and 75th percentiles, and whiskers extend to the most extreme data points not considered outliers. Outliers plotted individually.

3. The use of the CV procedures is problematic for univariate feature selection. It is important not to claim these as predictive features and the manuscript needs to be revised accordingly.

We thank the reviewer for pointing out this important point; the use of the word “feature” in previous versions of the manuscript was imprecise. Because models are trained and tested using network strength summary statistics (i.e., the sums of many edges’ strengths), the edges themselves are not predictive features. To clarify this point, all mentions of “feature selection” in the manuscript and SI have been changed to “edge selection.”

4. Interpretation of features remains problematic: the authors’ justification for using gambling, motor, social and emotion tasks to predict fluid intelligence is still weak, and this aspect has not been improved. The authors have not discussed what the features mean and how they contribute to fluid intelligence. Instead, they note that it is unsurprising that e.g. motor task-based models perform relatively well. Similarly, explaining fluid intelligence from gambling task features is not quite meaningful or interpretable. If the goal is solely prediction and the features are essentially uninterpretable, why discuss the features and tasks at such length?

The primary goal of this work is to demonstrate that cognitive tasks amplify individual differences in neural circuitry that are related to individual differences in behavior and cognition. Prediction is used as a means to demonstrate this point; that task-based models outperform rest-based models indicates that task-based functional connectivity patterns contain more information relevant to such individual differences than rest-based functional connectivity patterns. Given the stability of model anatomical distribution (see response to Reviewer 2, comment 2), we analyze this anatomy to offer some insight into which gF-relevant edges are detectable in various brain states. We note again that the broad spatial distribution of selected edges highlights the importance of a whole-brain, data-driven approach to studying the neural bases of such complex cognitive constructs. Our goal is not to map a complete fluid intelligence network, nor to explain why tasks differentially improve gF prediction, though we agree that these are important and interesting questions for future work.

Reviewer #3 (Remarks to the Author):

The authors have addressed all of my comments thoroughly. I have no further suggestions.

References

1. Yarkoni, T. & Westfall, J. Choosing prediction over explanation in psychology: lessons from machine learning. *Perspect. Psychol. Sci.* **12**, 1100–1122 (2017).
2. Turk-Browne, N. B. Functional interactions as big data in the human brain. *Science* **342**, 580–584 (2013).
3. Power, J. D., Schlaggar, B. L. & Petersen, S. E. Studying brain organization via spontaneous fMRI signal. *Neuron* **84**, 681–696 (2014).

REVIEWERS' COMMENTS:

Reviewer #1 (Remarks to the Author):

The authors have been very responsive, and I am satisfied with the revised version.

Reviewer #2 (Remarks to the Author):

The authors have done an excellent job of addressing the issues raised.